

# Debris cover and the thinning of Kennicott Glacier, Alaska, Part C: feedbacks between melt, ice dynamics, and surface processes

Leif S. Anderson[1,2], William H. Armstrong[1,3], Robert S. Anderson[1], and Pascal Buri[4]

[1]Department of Geological Sciences and Institute of Arctic and Alpine Research, University of Colorado, Boulder, CO, USA

[2]GFZ German Research Centre for Geosciences, Potsdam, Germany

[3]Department of Geological and Environmental Sciences, Appalachian State University, Boone, NC,

USA

[4]Geophysical Institute, University of Alaska-Fairbanks, Fairbanks, AK, USA

*Correspondence to*: Leif Anderson (leif@gfz-potsdam.de)

**Abstract.** The mass balance of many valley glaciers is enhanced by the presence of melt hotspots within otherwise continuous debris cover. We assess the effect of debris, melt hotspots, and ice dynamics on the thinning of Kennicott Glacier in three companion papers. In Part A we report in situ measurements from the debris-covered tongue. In Part B, we develop a method to delineate ice cliffs using high-resolution imagery and produce distributed mass balance estimates. Here in Part C we describe feedbacks controlling rapid thinning under thick debris.

Despite the extreme abundance of ice cliffs on Kennicott Glacier, average melt rates are strongly suppressed downglacier due to thick debris. The estimated melt pattern therefore appears to reflect Østrem's curve (the debris thickness-melt relationship).

As Kennicott Glacier has thinned over the last century Østrem's curve has manifested itself in two process domains on the glacier surface. The portion of the glacier affected by the upper-limb of Østrem's curve corresponds to high melt, melt

gradients, and ice dynamics, as well as high ice cliff and stream occurrence. The portion of the glacier affected by the lower-limb of Østrem's curve corresponds to low melt, melt gradients, and ice dynamics, as well as high ice cliff and stream occurrence.

The upglacier end of the zone of maximum thinning on Kennicott Glacier occurs at the boundary between these process domains and the bend in Østrem's curve. The expansion of debris upglacier appears to be linked to changes in the surface

velocity pattern through time. In response to climate warming, debris itself may therefore control where rapid thinning occurs on debris-covered glaciers. Ice cliffs are most abundant at the upglacier end of the zone of maximum thinning.



**Keywords:** ice cliff, stream, lake, velocity, Østrem's curve

**1 Introduction.**

Thick debris on glaciers insulates and strongly reduces surface melt. Despite this many debris-covered glaciers are thinning rapidly under thick debris. This contradiction is often referred to as the 'debris-cover anomaly' (Pellicciotti et al., 2015). The two potential explanations for the debris-cover anomaly are 1) *melt hotspots* within otherwise thick debris or 2) changes in ice dynamics. *Melt hotspots* are associated with surface features like ice cliffs or lakes within otherwise continuous debris cover. Changes in ice dynamics are most notably manifested in the reduction of ice flow to the debris-covered portion of the glacier.

For debris-free glaciers a feedback between increased surface melt and reduced ice flow down valley can explain the typical pattern of thinning (Nye, 1960). We should therefore also expect feedbacks to define the thinning pattern of debris-covered glaciers. Recent work has shown that both melt and declining ice flow play a role in thinning the debris-covered Changri Nup Glacier in Nepal (Vincent et al., 2016; Brun et al., 2018). Melt and ice dynamics also appear to control the location of the zone of maximum thinning (*ZMT*) at Kennicott Glacier, Alaska (Fig. 1; Part B).

Both surface melt and ice dynamics are fundamental for explaining the thinning of debris-free and debris-covered glaciers alike. The continuity equation for ice is the key to understanding glacier thinning:

$$\frac{dH}{dt} = \dot{b} - \frac{dQ}{dx} - \frac{dQ}{dy} \quad , \qquad (1)$$

where $H$ is the ice thickness, $t$ is time, $\dot{b}$ is the annual specific ablation (or loosely ice melt in the ablation zone), and $Q$ is the column integrated ice discharge. The last two terms on the right combine to represent ice emergence or equivalently surface uplift. These last two terms loosely represent ice dynamics.

Debris-covered glaciers add an additional component that is not explicitly referenced in Eq. (1): surface processes. These surface processes help set the distribution of ice cliffs, lakes, and streams that in turn feedback to set the melt rate and pattern of ice dynamics. But how these surface features and processes feedback into melt and ice dynamics is largely unknown. To determine the specific process that cause rapid thinning under thick debris we must identify the most important feedbacks between melt, ice dynamics, and surface processes. Ultimately quantifying the most vital feedbacks will facilitate the prediction of debris-covered glacier response to climate change and potentially allow for the simple explanation of the debris-cover anomaly.

In Part C we therefore address: *What process links and feedbacks are occurring on Kennicott Glacier that contribute to glacier thinning under thick debris?* Identifying these feedbacks requires that each of the three fundamental components be quantified across debris-covered glaciers. In Parts A and B we quantify patterns of debris, melt, as well as the distribution of ice cliffs and lakes across the debris-covered tongue of Kennicott Glacier. We show that the melt pattern through the study area likely follows the shape of Østrem's curve downglacier (Figs. 2 and 3). Here in Part C, we present estimates of ice dynamics and supraglacial streams across the study area.

**1.1 Study glacier**





Kennicott Glacier is a broadly south facing glacier located in the Wrangell Mountains of Alaska (Fig. 1; 42 km long; 387 km$^2$ area). As of 2015, 20% of Kennicott Glacier was debris-covered. Below the equilibrium-line altitude at about 1500 m (Armstrong et al., 2017), 11 medial moraines can be identified on the glacier surface. Above 700 m elevation debris is typically less than 5 cm thick, although, locally, areas with low surface velocities tend to have higher debris thicknesses (Anderson and Anderson, 2018). Medial moraines coalesce 7 km from the terminus to form a debris mantle with ice cliffs,

streams, and lakes scattered within an otherwise continuous debris cover. Kennicott Glacier supports more ice cliffs per area than any other studied debris-covered glacier and surface lakes are abundant near the terminus (see Part B as well as Rickman and Rosenkrans,1997).

Kennicott Glacier has been the focus of outburst flood and ice dynamics research for almost two decades. Each year the ice-marginal Hidden Creek Lake drains under Kennicott Glacier (Rickman and Rosenkrans, 1997; Anderson et al., 2003a,

2003b; Walder et al., 2005, 2006). The outburst flood increases basal water pressures, leading to a 1-2 day period of enhanced basal sliding (Anderson et al., 2005; Bartholomaus et al., 2008, 2011; Armstrong et al., 2016). Armstrong et al. (2016, 2017) showed that a significant portion of surface displacement in the debris-covered tongue of Kennicott Glacier occurs in the summer, due to sliding. In the lowest 4 km of the glacier almost all of the motion is accomplished by sliding and/or longitudinal coupling to the actively deforming ice immediately upglacier.

Rickman and Rosenkrans (1997) describe a topographic bulge (rapid increase in surface slope) on the glacier surface 4 km upglacier from the terminus. The velocity maps of Armstrong et al. (2016) show a rapid increase in surface velocity at a similar location. Active, deforming ice was likely near the modern terminus until 1948, but by 1990 the lower 4 km of the glacier had thinned significantly lowering surface slopes (Rickman and Rosenkrans, 1997).

## 2 Methods

### 2.1 Annual surface velocities

Annual surface velocities were estimated using GoLIVE velocity maps between 2013 and 2018 (Fahnestock et al., 2016; Scambos et al., 2016). In order to estimate annual surface velocities we first calculated average daily summer and winter velocities based off of 37 available velocity maps derived from image pairs greater than 60 days apart. We used mask and threshold values twice as stringent as those provided in the GoLIVE output. Velocity maps were differentiated into summer

(May 1$^{st}$ to September 30$^{th}$) and winter categories based on when air temperatures are above 0°C (after Armstrong et al., 2017). Total mean annual summer and winter displacements were then calculated and summed to estimate annual average surface velocities. The surface velocity patterns we estimate here are similar to those from Armstrong et al. (2016) and Armstrong et al. (2017) based on WorldView and Landsat imagery. New analyses were required to estimate the annual velocity pattern.

### 2.2 Ice emergence rate

Ice emergence rates were calculated based on the observation that ice discharge, $Q$ in Eq. (1) can be rewritten in terms of ice thickness $H$ and the column averaged downglacier velocity $\bar{U}$ :

$$Q = \bar{U} H \quad . \quad (2)$$





Ice emergence rates were then calculated using:


$$w = \frac{-\partial \bar{U}_x H}{\partial x} - \frac{\partial \bar{U}_y H}{\partial y} \quad (3)$$

where $\bar{U}_x$ and $\bar{U}_y$ are the ice-column averaged velocities in the downglacier ($x$) and cross-glacier ($y$) directions, respectively, and $H$ is the ice thickness. We assume that winter velocities are solely caused by internal deformation and that increased velocities in the summer are caused by basal sliding. We corrected the surface velocity due to internal deformation to the column-mean velocity by multiplying the deformation-dependent surface velocities by 4/5 (after

Anderson and Anderson, 2010). The summer-sliding velocities were then added to the corrected-deformation velocities to estimate the column-mean velocity across the study area. We acknowledge that winter sliding may exist (e.g., Raymond, 1971; Amundson et al., 2006; Armstrong et al., 2016), but its magnitude cannot be quantified without borehole observations or better knowledge of the ice thickness distribution. In the unlikely case that winter motion is driven solely by basal sliding, the surface velocity would equal the column-average velocity and the estimate of emergence rate would increase by

20%. Ice thickness estimates were derived from Huss and Farinotti (2012). The maximum estimated ice thickness in the study area is 120 m (Supplemental Figure 1). We also estimate ice emergence rates assuming a uniform bed under the glacier fixed at the terminus elevation.

**2.3 Digitization of supraglacial streams**

To quantify the extent of surface streams we hand-digitized streams from the 2009 WV summer image in QGIS. Streams were searched for in a grid across the study area. Supraglacial stream sinuosity $S$ was calculated using:

$$S = \frac{L_{channel}}{L_{straight}} \quad (4)$$

where $L_{channel}$ is the length of the channel and $L_{straight}$ is the straight-line distance downstream. A higher value of $S$ corresponds to a more sinuous stream. Each $S$ calculation was performed for linear stream reaches. Nodes of each stream path were also

resampled to 10 m segments, allowing for the estimation of stream length per area.

**3. Results**
**3.1 Annual surface velocities and ice emergence rates**

Maximum annual surface velocities (75 m yr[-1]) occur in the upper portion of the study area near the center of the glacier

(Fig. 4). Surface velocities tend to decrease downglacier and towards the glacier margin. Ice free error check areas, (totaling 4.5 km[2]) adjacent to the glacier and below 500 m produce mean annual velocities in the x- and y-directions of -6.3 and 2.4 m yr[-1] respectively. We therefore take the uncertainty of the surface velocity estimates to be ± 6.3 m yr[-1]. Rapid changes in the decline of surface velocities (i.e., longitudinal strain rate) are also observed ~ 4 km from the terminus (Fig. 5c). Maximum ice emergence rates occur ~ 4 km from the terminus for both the variable bed and uniform bed cases (Fig. 5d).

Ice emergence rates also decline to zero in the lowest 4 km of the glacier for both the variable bed and uniform bed cases. The maximum estimated decline of surface velocities and emergence rates coincide with the topographic bulge in the 2009 surface profile (Fig. 5e).





### 3.2 Supraglacial streams and sinuosities

Supraglacial streams show a bi-modal pattern across the study area. Streams are abundant where debris is thinner than ~20 cm and almost absent where debris is thicker than ~20 cm (Fig. 5g). At the transition between these domains, streams descend into a series of moulins running across the glacier (Fig. 6). Other streams flow off the edge of the glacier and join ice-marginal rivers. Stream sinuosities were low in the upper portion of the study area and increased downglacier until ~ 4 km from the terminus where debris is about 20 cm thick (Fig. 7).


## 4 Discussion

### 4.1 The primacy of Østrem's curve

Two observations are key to the discussion of patterns and process links that follows. First, debris tends to thicken downglacier (Figs. 3 and 5a; e.g., Rounce and McKinney, 2014; Anderson and Anderson, 2018). Second, the melt pattern

downglacier tends to follow the debris thickness-melt relationship or Østrem's curve (Figs. 2,3, and 5b). The patterns and feedbacks occurring on Kennicott Glacier can be broken into two process domains (Fig. 3): 1) the portion of the glacier affected by the upper-limb of Østrem's curve; and 2) the portion of the glacier affected by the lower-limb of Østrem's curve. Here, we divide Østrem's curve into limbs based on a threshold of 20 cm debris thickness, but in principle the division could be made at any debris thickness between 10 and 20 cm (Fig. 2). In the portions of the glacier affected by the

upper limb of Østrem's curve, small differences in debris thickness lead to large differences in melt rate (Figs. 2 and 5). The low-melt and low-gradient 'lower-limb' of the debris-perturbed mass balance profile tends to be extended compared the 'upper-limb' portion (see Fig. 5 from Anderson and Anderson, 2016). We start by discussing the melt profile and incrementally add ice dynamics, surface processes, and finally the pattern of thinning.

### 4.1.1 Manifestation of Østrem's curve in the mass balance profile

For Kennicott Glacier, debris tends to monotonically thicken downglacier, as is typical for other debris-covered glaciers (e.g., Anderson and Anderson, 2018). This leads to the expectation that sub-debris melt rates should also decline towards the terminus despite increasing available energy at lower elevations. At least on Kennicott Glacier, *melt hotspots* (ice cliffs specifically) do not increase area-averaged melt rates nearly enough to compensate for the the melt-suppressing effects of thick debris (Fig. 3). This inference is based on abundant in situ measurements (Part A), ice cliff extents and distributed

melt estimates (Part B). Average melt rates decline in a non-linear fashion from the top of the study area towards the terminus, broadly following Østrem's curve (Fig. 5).

### 4.1.2 Manifestation of Østrem's curve in the pattern of ice dynamics

Figure 5 shows how the upper- and lower-limbs of Østrem's curve are imprinted in the pattern of ice dynamics at Kennicott

Glacier. Annual surface velocities correlate with the inverse of debris thickness and melt rate. This follows the theoretical explanations for steady state debris thickness patterns where debris emergence rates are small, as they are expected to be



under thick debris (Anderson and Anderson, 2018). Glacier surface mass balance inherently feeds back into surface velocities and ice fluxes. In steady state (the attractor state for the system) Eq. (1) becomes:

$$\dot{b} = \frac{\partial Q}{\partial x} + \frac{\partial Q}{\partial y} \quad , \quad (5)$$

in the ablation zone the left side of the equation is ice melt and the right side of the equation is ice emergence. On debris-covered glaciers debris thickness controls melt rate which is also controlled by ice dynamics. This can be seen in the continuity equation for debris thickness $h_{debris}$ (e.g., Nakawo et al., 1986):

$$\frac{\partial h_{debris}}{\partial t} = \frac{C\,\dot{b}}{(1-\phi)\rho_r} - \frac{\partial(U_x^s h_{debris})}{\partial x} - \frac{\partial(U_y^s h_{debris})}{\partial y} \quad (6)$$

where $C$ is the englacial debris concentration, $\varphi$ is the porosity of the debris, $\rho_r$ is the density of the rock composing the
debris, and $U^s$ is surface velocity in the $x$- and $y$- directions. The first term on the right hand side of Eqn. (6) is the debris emergence rate and the second two terms are the effect of ice dynamics on debris thickness. Based on the theory presented here (and further elaborated on in Anderson and Anderson (2016; 2018) debris thickness, melt rates, and surface velocity patterns should all complement one another (also see Rowan et al., 2015; Watson et al., 2017). Where debris is thin and melt rates are high ice flow should also be high. Where debris is thick and melt rates are low ice flow should be low.


On Kennicott Glacier debris, melt rates and velocities appear to follow these theoretical expectations (Fig. 5, Eqs. 5 and 6). Where debris is thin (< 20 cm) melt rates are high, surface slope, velocities, strain rates, and ice emergence rates also tend to be high. In the lower portion of the glacier where debris is thick, melt rate and melt rate gradients are low as are surface slopes, velocities, strain rates, and ice emergence rates.


Theory points to surface debris and ice dynamics co-evolving in a 'chicken' or 'egg' fashion. Causality in some cases may be difficult to assign: debris thickness and surface velocity feedback to control one another (see Eqs. 5 and 6). If surface velocities are strongly controlled by bed topography then debris thickness will evolve to match the ice dynamical control. If debris is continuous and thick it will reduce glacier surface slope and flow. Most importantly, observed debris thickness,
melt rate, and ice dynamics patterns appear to be self-consistent on Kennicott Glacier. And this self consistency appears to occur for other debris-covered glaciers as well (e.g., Thompson et al., 2016; Watson et al., 2017).

### 4.1.3 Manifestation of Østrem's curve in the pattern of surface features

Ice cliffs are exceptionally abundant on Kennicott Glacier but they are also distributed in a bi-modal fashion (Figs. 5 and 8).
The 'upper limb' portion of the study area has an exceptional ice cliff fractional area of 12.4 % and the 'lower limb' portion has an ice cliff fractional area of 8.5 %. This 8.5 % ice cliff fractional area is similar to coverage percentages observed on other debris-covered glaciers (see Part B). Streams show an even stronger bi-modal pattern. Streams are abundant in the 'upper limb' portion and almost absent in the 'lower limb' portion of the study area (Fig. 5g). Lakes show the opposite pattern and are less abundant in the 'upper limb' portion and more abundant in the 'lower limb' portion of the study area





(Fig. 5g). There are a number of process links and feedbacks that link the debris, ice dynamics and surface features. We identify and describe some here.

*Østrem's curve controls ice cliff distribution.* In the portion of Kennicott glacier affected by the upper limb of Østrem's curve, small differences in debris thickness lead to large differences in melt rate (Figs. 2 and 5). Spatial differences in debris thickness (over the 1-meter to 100-meter scale) can thus lead to the creation of relief and locally steepened slopes. Small-

scale surface topography allows for debris layer mass wasting (e.g., Moore, 2018), failure at the debris-ice interface, and ice cliff nucleation. In the portion of the glacier affected by the lower limb of Østrem's curve, differences in debris thickness lead to small differences in melt rate. This means that surface relief and slopes are unlikely to increase reducing the propensity of debris failure and ice cliff nucleation. Where debris is thick, ice cliffs are more likely to be buried and removed from the glacier surface (Fig. 9). We expect this mechanism to occur on other debris covered glaciers where debris

is less than about 20 cm thick.

*Active ice dynamics and ice cliffs.* In the portion of the glacier affected by the upper limb of Østrem's curve, emergence and strain rates are high, potentially disturbing and steepening local ice surfaces, increasing the chance for the mass wasting of surface debris and ice cliff nucleation. The highest concentration of ice cliffs correlates with the highest emergence and

strain rates in the study area (Fig. 5). During the summer, high surface strain rates occur where ice cliffs are most abundant (Armstrong et al., 2016) suggesting that the extreme ice cliff abundance on Kennicott Glacier may be related, to basal sliding and its effect on surface strain and ice emergence rates. In the Khumbu region of Nepal, Watson et al. (2017) documented a statistical correlation between ice cliff occurance and active flow, but not in the bi-modal fashion observed on Kennicott Glacier. The link between active flow and ice cliff distribution may be common on other debris-covered glaciers,

especially where debris is thin and ice cliffs are less likely to be buried.

*Surface streams maintain ice cliffs.* A positive feedback between ice cliffs and surface streams is apparent on Kennicott Glacier. Surface streams and ice cliffs are abundant where debris is thin (Fig. 5). High sub-debris melt and abundant, backwasting ice cliffs increase surface water availability. On the glacier, we observed many streams undercutting ice cliffs (Fig. 9). Streams create a gap between the ice cliff and the debris-covered surface, preventing ice cliff burial and

transporting debris downstream. Stream sinuosity also correlates with ice cliff abundance (Fig. 7). Streams tend to be more sinuous where ice surface slopes are steep and water discharge is high (e.g., Ferguson, 1973). Both stream sinuosity and ice cliff fractional area are maximized where glacier surface slopes are steepest and stream discharge is expected to be highest across the study area. If stream sinuosity correlates with propensity for stream meandering then the effect would be amplified and potentially leading to the formation englacial tunnels (e.g., Benn et al., 2017). These englacial tunnels could

then lead to the formation of thermokarst depressions downglacier. Sinuous streams tend to undercut and potentially cause arc shaped ice cliffs that in turn focus falling debris at their base (Fig. 9). This mechanism locally increases debris thicknesses.

The lowest 4 km of the glacier is hydrologically disconnected from the upper portion of the glacier (Fig. 6). Here, where

debris is thick, melt rates are low and ice cliffs are less abundant. Surface water discharge will be low as will the potential for sediment mobilization by streams. Ice cliffs are therefore more likely to be buried (Fig. 9). Thermokarst depressions also



dominate the glacier surface topography reducing drainage basin size and stream discharge (Figs. 5 and 6). Ultimately feedbacks between streams and ice cliffs will be most apparent where there are large areas of thin debris, though the influence of streams on ice cliffs has also been observed where debris is thicker than 20 cm (Miles et al., 2017).


*Thick debris, reduced ice flow and lakes.* The expansion of lakes upglacier between 1957 and 2009 coincides with a reduction of mean surface slope from 2.5° to 1.4° (see Reynolds, 2000 for similar observations related to surface slope from Nepal). On Kennicott Glacier, surface lakes are more abundant where debris is thick as well as where surface slopes and velocities are low (also see Quincey et al., 2007; Sakai and Fujita, 2010). Thick debris will help maintain thermokarst

basins. Differences in debris thickness, where average debris is thick will produce small melt rate gradients and small changes in topography. This effect will stabilize the glacier surface landscape. Where debris is thick, thermokarst basins are less likely to close due to active ice flow. Small surface velocity gradients will reduce crevasse formation and lake draining events.

We synthesize the process links described up to this point in Figure 10. This figure highlights how debris, ice dynamics, and surface features interact independent of the effects of climate change. Processes links directly related to climate warming and glacier thinning are described in the next section.

**4.2 Manifestation of Østrem's curve in the thinning pattern**

On Kennicott Glacier, predicted maximum, glacier-wide melt rates do not coincide with the zone of maximum thinning

(*ZMT*). This suggests that changes in ice flow help control the location of the *ZMT*. Surface velocities, and thus ice discharge are highly dependent on ice thickness and surface slope. Ice discharge scales with $H$ to the power of five and the surface slope to the power of three (e.g., Cuffey and Paterson, 2010). From the top of the glacier toward the terminus, local glacier thinning has a compounding, reducing effect on ice discharge independent of the presence of debris (Nye, 1960).

On Kennicott Glacier, the upglacier end of the *ZMT* coincides with the transition between the upper- and lower-limbs of Østrem's curve manifested in the mass balance pattern. In the portion of the glacier affected by the lower-limb of Østrem's curve the glacier has thinned and surface slopes have declined from 2.5° to 1.4° from 1957 to 2009. This is consistent with declining emergence rates through time. In this portion of the glacier, modern estimated emergence rates decline rapidly to zero (Fig. 5).

Meanwhile, the portion of the glacier in the upper-limb of Østrem's curve has thinned, and surface slopes have increased from 2° to 2.4° between 1957 and 2009. This is reflected in the formation of the topographic bulge at the upglacier end of the *ZMT* (Fig. 5; Rickman and Rosenkrans, 1997), where emergence and strain rates are highest. Any reduction in these high emergence rates at the upper end of the *ZMT* (e.g., due to the ongoing decline of ice discharge from upglacier) will lead to rapid thinning.

In the portion of the glacier affected by the upper-limb of Østrem's curve, where debris tends to be less than 20 cm thick, melt rates are higher, consistent with higher ice emergence rates (see Anderson and Anderson, 2016; Crump et al., 2017).





The location of the *ZMT* on Kennicott Glacier may therefore be related to the pattern of debris, changes in this pattern time, and the imposed effect of changing debris thickness on ice emergence rates.

### 4.2.1 Locally thick debris allows for debris expansion

Figure 11 shows the expansion of debris on the surface of Kennicott Glacier between 1957 and 2009. Over this period debris stripes on the glacier surface have been deformed in a subparallel fashion (Fig. 11). The consistent change in shape of these stripes suggests that the velocity field has changed considerably. As Kennicott Glacier has thinned the bed, valley wall, and ice surface topography has increasingly influenced flow. The surface flow field appears to have become less linear and more S-shaped through time. Changing flow directions allows debris to be transported into previously debris-free areas.

On Kennicott Glacier, debris expanded into valleys between medial moraines (Fig. 11). As glaciers thin, areas where debris is thicker (e.g., medial moraines) will tend to increase in relief potentially affecting ice flow and expanding debris. Changes in surface flow may be important for debris expansion and thickening on other glaciers (e.g., Kirkbride, 1993; Deline, 2005; Azzoni et al., 2018), especially where medial moraines are present.

### 4.2.2 Active flow increases ice cliff abundance and thinning

The thickening of debris down glacier leads to the control of Østrem's curve on melt rate and ice dynamics. Ice cliff fractional area is highest at the junction between the limbs of Østrem's curve, i.e., at the upper edge of the *ZMT*, and where strain and emergence rates are also highest (Fig. 5). This suggests that thin debris and active ice flow leads to increased ice cliff abundance, leading to increased melt rates. Higher melt rates will increase glacier thinning locally, propagating low surface slopes upglacier. While our distributed melt estimates (Part B) suggest that ice cliffs are not able to compensate for the insulating effects of debris, this feedback should enhance glacier thinning. This ice cliff-ice dynamics feedback will likely occur on other glaciers where ice cliff abundance is positively correlated with active flow, as has been previously documented for the Khumbu region of Nepal (Watson et al., 2017).

### 5 Conclusions

On Kennicott Glacier, debris thickens downglacier (Part A), leading to the imprinting of Østrem's curve on the melt pattern (Part B). The upper- and lower-limbs of Østrem's curve manifest themselves in two process domains on the glacier surface. Where debris is thinner than 10-20 cm melt rates, melt rate gradients, and ice flow are high. Ice cliffs and streams are abundant, but lakes are largely absent. Where debris is thicker than 10-20 cm melt rates, melt rate gradients, and ice flow are low. Ice cliffs and streams are less abundant, but lakes are common. We propose that these two process domains will also be present on other debris-covered glaciers as well.

On Kennicott Glacier, debris, melt, and ice dynamics patterns are consistent with established theory of debris-covered glaciers (Anderson and Anderson, 2016; 2018). Following that theory and observations from Kennicott Glacier, debris thickness itself helps define the pattern of ice emergence rates. The upglacier end of the zone of maximum thinning (*ZMT*)



coincides with 10 to 20 cm debris thicknesses and a bend in Østrem's curve. Debris expansion over the last 70 years appears to be related to changes in the surface velocity pattern. This points to a feedback between changes in the debris pattern, changes in ice emergence rates, and climate warming at Kennicott Glacier. An exceptional abundance of ice cliffs correlates

with the upglacier end of the *ZMT*. On Kennicott Glaicer ice cliffs do not backwaste fast enough to compensate for the melt insulating effects of thick debris (Part B). But an ice cliff-glacier thinning feedback is evident on Kennicott Glacier and potentially on other glaciers where active flow and ice cliff abundance are positively correlated.

**Data availability**

Datasets and results are available upon request.

**Author contribution statement**

LSA designed the study, composed the manuscript, and all analyses. WHA helped develop the annual surface velocities. RSA advised LSA and WHA through the study and contributed to the text. PB and DS added important discussion points that improved the manuscript. All authors revised the manuscript.

**Competing Interests**

The authors declare that they have no conflict of interest.

**Acknowledgements**

LSA acknowledges support from a 2011 Muire Science and Learning Center Fellowship and NSF DGE-1144083 (GRFP). RSA and WHA acknowledge support of NSF EAR-1239281 (Boulder Creek CZO) and NSF EAR-1123855. WHA acknowledges support from NSF OPP-1821002 and the University of Colorado at Boulder's Earth Lab initiative. We thank

Craig Anderson, Emily Longano, and Oren Leibson for field support. We thank Per Jenssen, Susan Fison, Ben Hudson, Patrick Tomco, Rommel Zulueta, the Wrangell-St. Elias Interpretive Rangers, the Wrangell Mountains Center, Indrani Das, and Ted Scambos (NSIDC) for logistical support and the gracious loan of equipment. We thank Lucy Tyrell for facilitating outreach efforts. We also thank Joshua Scott, Wrangell-St Elias National Park and the Polar Geospatial Center for access to satellite imagery. We thank Regine Hock, Martin Truffer, Evan Miles as well as the research groups of both Andreas Vieli

and Francesca Pellicciotti for helpful discussions. LSA thanks the organizers and participants of the 2010 Glaciological Summer School held in McCarthy, AK, which inspired this work.

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



**Figures**

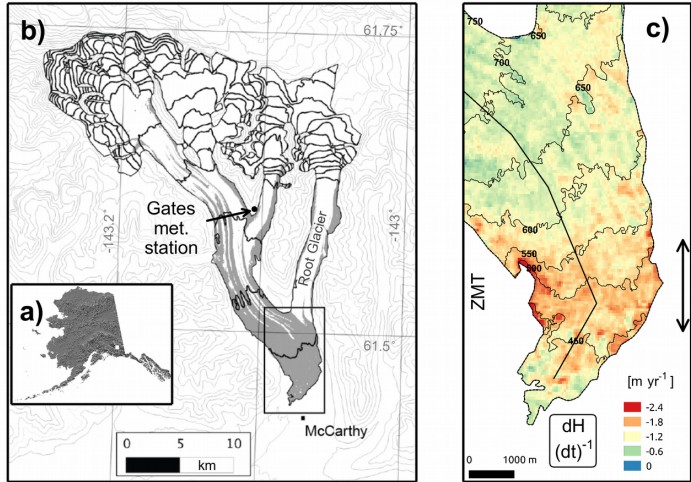

**Figure 1. Context map and the pattern of thinning within the study area.** a) Map of Alaska showing the location of panel b and the Wrangell Mountains. b) Kennicott Glacier with the location of the Gates glacier meteorological station (1240 m a.s.l.). c) Map of the study area (24.2 km$^2$) with the mean glacier surface lowering rate (dH (dt)$^{-1}$) between 1957 and 2009 (see Das et al., 2014) (mean error 0.04 m yr$^{-1}$ and 1std 0.15 m yr$^{-1}$ based on 3 km$^2$ area near the modern terminus). The maximum plausible error based on reported uncertainties from the digital elevation models is 0.58 m yr$^{-1}$ which only occurs when each DEM has an extreme error with the opposite sight in the same pixel. *ZMT* refers to the zone of maximum thinning. The profile used to present data with distance from the terminus in later figures. Swath profiles presented lower are 1000 m wide.

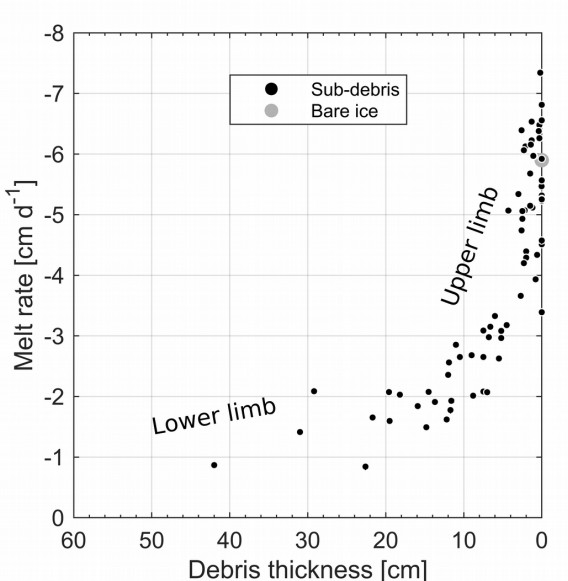

**Figure 2. The debris thickness-melt relationship or Østrem's curve from Kennicott Glacier.** Data collection described

in Part A. The division between the upper- and lower-limbs of the curve is arbitrary and could be defined anywhere between

10 and 20 cm.




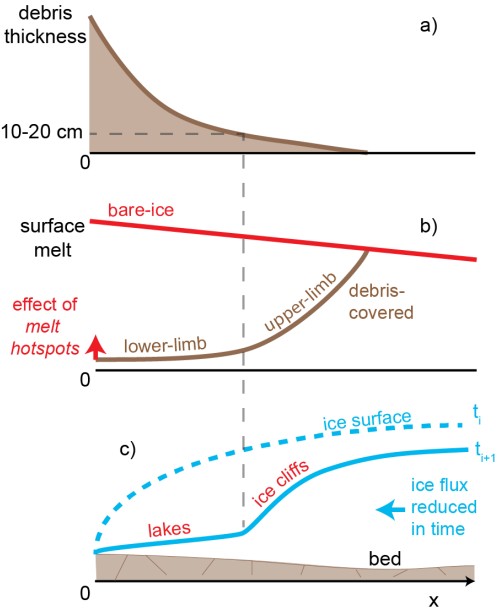

**Figure 3. Schematic showing the idealized thinning of Kennicott Glacier.** a) Idealized debris thickness pattern. b) Surface melt rate. The bare-ice melt rate increases down glacier as available energy increases downglacier. Because debris thickens downglacier the insulating effects of debris cover become stronger (in brown). The sub-debris melt pattern is expected to reflect Østrem's curve. Melt hotspots (like ice cliffs, lakes, and streams) will locally increase the melt rate and average melt rates towards bare-ice melt rates (Part B). c) Idealized representation of the thinning of Kennicott Glacier. The $t_i$ surface profile represents the surface profile at the end of the Little Ice Age. The $t_{i+1}$ surface profile represents the surface profile observed today. More lakes are observed in the lower-limb portion of the glacier and more ice cliffs are present in the upper-limb portion of the glacier.









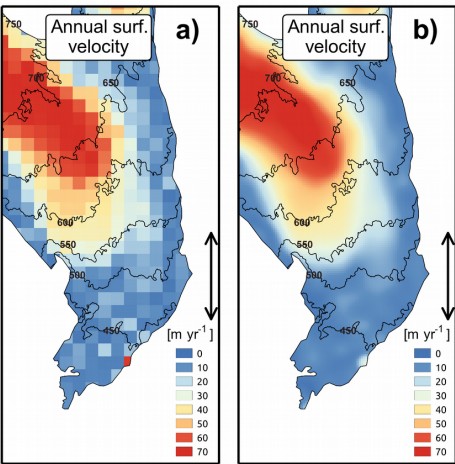

Figure 4. Annual surface velocities. Velocities are derived from GoLIVE output (Fahnestock et al., 2016; Scambos et al., 2016). We estimate the uncertainty of the measurements to be ± 6.3 m yr$^{-1}$. The zone of maximum thinning (*ZMT*) (see Fig. 1c) is referenced by the double headed arrow in each panel. a) The average annual surface velocities (between 2013 and 2018) at 300 m resolution. b) Same data from a) but smoothed with a Gaussian filter. The box shows the extent of Figure 11.


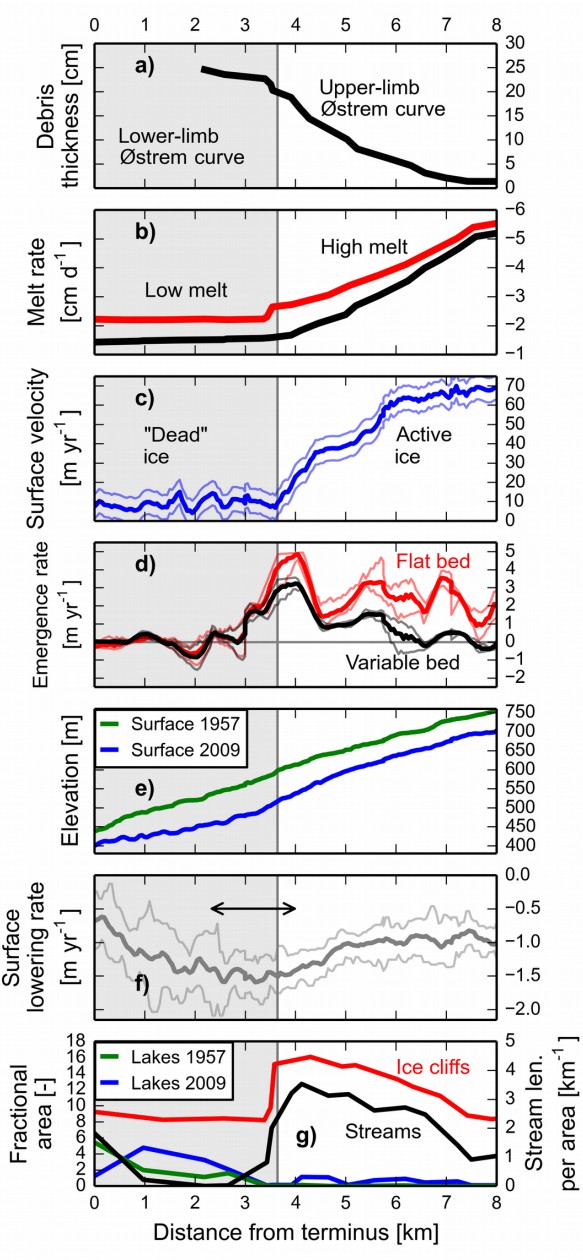

**Figure 5. Comparison of glacier properties near the terminus of Kennicott Glacier.** a) Debris thickness with distance from the terminus (Part A). No debris thickness measurements were made in the lower 2 kilometers of the glacier. b) Distributed melt rate estimates from just sub-debris melt (black) and the combined melt from sub-debris and ice cliff melt (red) (see Part B for uncertainty estimates). c) Average annual surface velocity from 2013-2018. Thin lines are the extreme error bounds ± 6.3 m yr⁻¹. d) Thin lines are the maximum and minimum velocities within the swath profile. The variable bed






case uses the ice thickness map based on Huss and Farinotti (2012). The flat bed case uses a bed elevation fixed at 375 m a.s.l. e) Glacier surface elevation from 1957 (green) and 2009 (blue). Both DEMs have an uncertainty of ±15 m (Das et al., 2014). f) Surface lowering rate between 1957 and 2009. Thin lines are the maximum and minimum values within the swath profile. Where surface velocities and emergence rate is low the glacier surface is more variable. The double-headed arrow

represents the zone of maximum thinning (*ZMT*) referenced elsewhere. g) The fractional coverage of ice cliffs, lakes, and streams. As described in Part B ice cliff coverage has an extreme uncertainty of ± 20%.









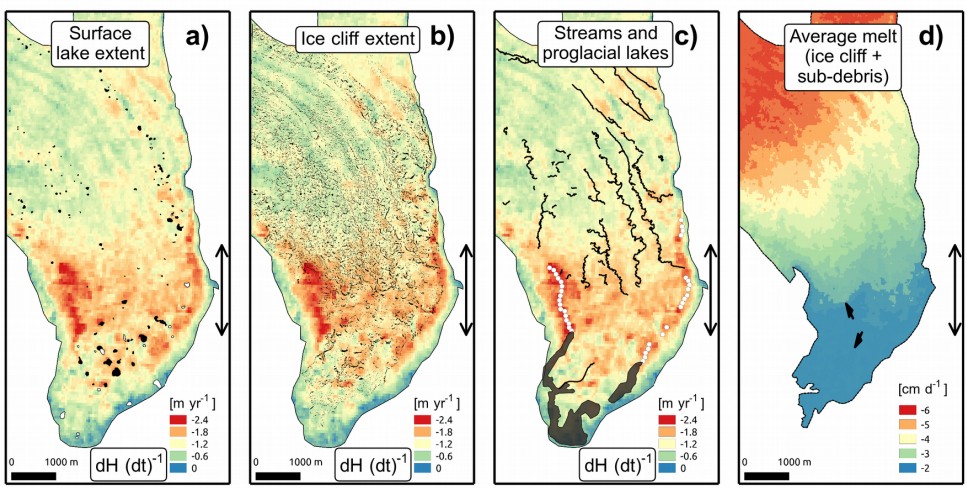

**Figure 6. Comparison between glacier surface lowering (dH (dt)$^{-1}$ from 1957 to 2009) and other glaciological properties.** The arrows indicate the extent of the zone of maximum thinning or (*ZMT*). a) Glacier surface lowering and surface lake extent. Observed surface lakes from the 1957 aerial photo are shown with white fill and black outlines. Observed surface lakes from 2009 from WV imagery are shown in black. b) Glacier surface lowering and 2009 ice cliff extent using an adaptive binary threshold (Part B). c) Supraglacial streams digitized from the 2009 WorldView image are shown as black lines. Ice marginal streams digitized from the 2009 WorldView image are show with white marker strings. The proglacial lake is shown by the dark shaded area. d) Elevation-band averaged melt due to ice cliffs and sub-debris melt from the summer of 2011. The arrows on the glacier show the look direction of the photographs in Fig. 8. The arrow pointing downglacier corresponds to panel 8a. The arrow pointing upglacier corresponds to panel 8b.



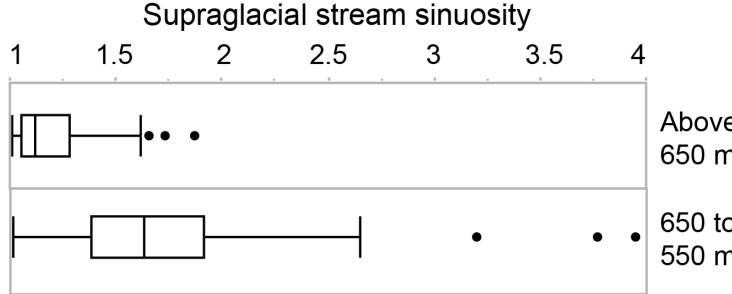

**Figure 7. Supraglacial stream sinuosity.** Stream sinuosity was determined based on the digitized supraglacial streams from Figure 6. Stream sinuosity of 1 represents a perfectly straight stream. Note that below 500 m elevation streams almost completely disappear. Outliers are shown as dots. The difference in sinuosity is also visible in Figure 6.









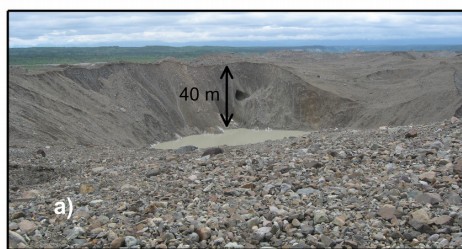 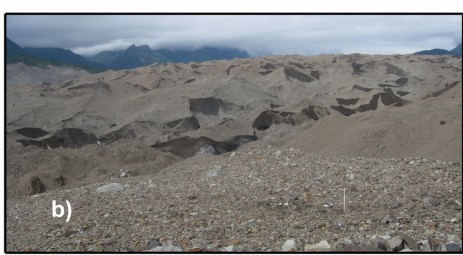

**Figure 8. Photos of the glacier surface at the location between the limbs of Østrem's curve expressed in the melt profile.** a) Portion of the glacier affected by the lower-limb of Ostrem's curve with abundant lakes and thermokarst depressions but relatively few ice cliffs. b) Looking upglacier towards the portion of the glacier affected by the upper-limb of Østrem's curve. This is also a photo of the topographic bulge referenced throughout the text. Ice cliffs are abundant but

lakes are rare. The ablation stake in the foreground is approximately 1 m tall. Photo look angles and locations are shown in Fig. 6d.





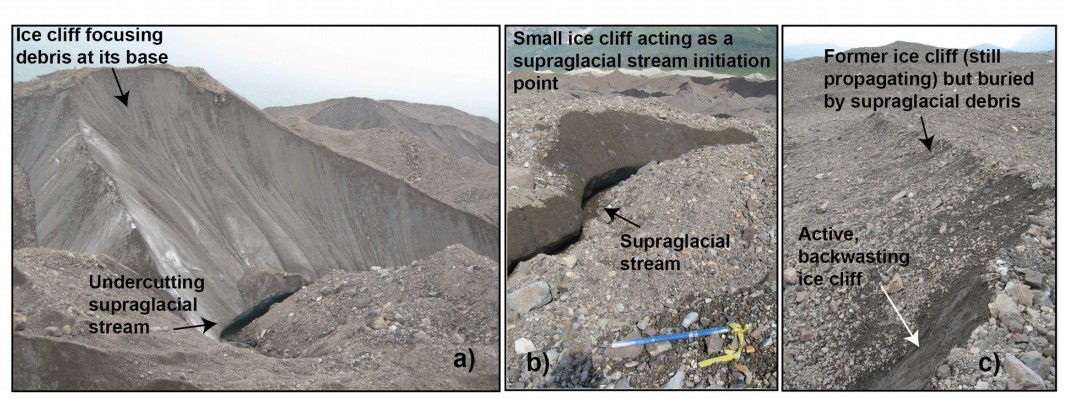

**Figure 9. Surface process links between debris, ice cliffs, and streams.** a) Approximately 10 m tall ice cliff. A sinuous

stream is undercutting the ice cliff. Drainage basins on some ice cliffs focus debris at their base (also see Watson et al.,

2017). Debris is noticeably thicker at the base of the ice cliff than at the top. b) A small ice cliff acting as the initiation point

of a stream. c) The ongoing burial of a small ice cliff in the portion of the glacier affected by the lower-limb of Østrem's

curve. The ice cliff at the lower edge of the image is approximately 0.5 m tall.





**Figure 10. Summary of expected relationship between debris, ice dynamics, and surface features.** Text in red points to a positive effect and text in blue points to a negative effect. Grey shading indicates that that the relationship would apply to the portion of the glacier affected by the lower-limb of Østrem's curve. The '—' represents an unknown or uncertain relationship. In summarizing the various relationships we try to describe the essence of the relationship in the table. The specific interaction between the two elements will certainty be more complicated and is expanded upon in the text. Feedback affects on thinning are described in the text.



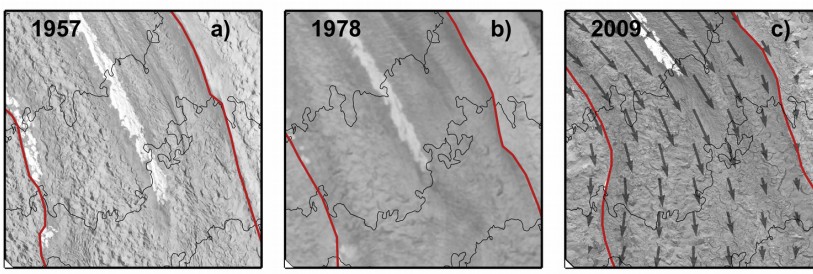

**Figure 11. Changes in debris extent through time on Kennicott Glacier.** Each panel reflects the same georefrenced area
that is 3 km by 3 km. Red lines represent the boarder between the same debris stripes in each panel. Note the distinct change
in the debris-stripe boundaries through time, especially between 1978 and 2009. a) 1957 aerial photo. b) 1979 aerial photo.
c) 2009 WorldView image with annual annual surface velocity vectors (from 2013 to 2018). The largest vector at the top of
the panel has a velocity of 75 m yr$^{-1}$. Gridded magnitudes of velocity are shown in Fig. 4. Note that flow is compressive
where debris extent has expanded. And that the flow direction parallels the digitized boundary between the debris stripes.