# Peer review of "Debris cover and the thinning of Kennicott Glacier, Alaska, Part C: feedbacks between melt, ice dynamics, and surface processes"

_The Cryosphere, 2019_

## Referee Comment (RC1) · David Rounce (Referee) · 16 Oct 2019

**Review of "Debris cover and the thinning of Kennicott Glacier, Alaska, Part C: feedbacks between melt, ice dynamics, and surface processes" by Anderson et al.**

This study is the third part of three publications that investigate debris cover on Kennicott Glacier in Alaska. The focus of this study is on the feedbacks between the melt, ice dynamics, and surface processes for the debris-covered portion of Kennicott Glacier. After reading all three parts, the introduction feels repetitive of Parts A and B (I recognize this is unavoidable). Unfortunately, the methods and results in this section are highly underwhelming. The new results in this part are surface velocities, emergence velocities based on those surface velocities and ice thickness, and manual delineation of streams for a single WorldView scene. These are all quite straightforward analyses that are fairly easy to perform.

Where this paper excels is in its discussion, which is grounded in the observations and results from Parts A and B with a little support from results and theory in Part C. While the discussion does not provide any conclusions that are necessarily groundbreaking (the relationship between debris thickness, surface velocities, and surface processes have been detailed for debris-covered glaciers in other parts of the world in other studies, which this manuscript references), it does provide an excellent holistic view of how the various feedbacks are connected for Kennicott Glacier and attempts to make universal statements concerning all debris-covered glaciers at times. The manuscript is well written, the figures support the text well, and there are sufficient references to the existing literature. Hence, my comments are fairly minor, but I admittedly have mixed feelings concerning the originality of the paper to stand on its own. If Parts A and B were separate studies by other authors, then I would argue that the originality and methodology would be poor-fair; this paper would come across more as a review paper of how existing studies are connected and likely not warrant publication without major revisions. However, that is not the case, and instead this paper comes across as an extension of Parts A and B, and a place where everything can be discussed in a broader context.

What would truly elevate this paper to stand on its own, would be if the theoretical feedbacks were supported by model results. Given that Part B develops empirical equations for accounting for distributed melting due to debris, ice cliffs and backwasting, this would be a very logical next step. However, I recognize that more information is likely needed concerning ice cliff nucleation and debris redistribution to be able to model these various surface processes over long periods of time and at a high enough level to support the discussion.

My recommendation would be to integrate Parts B and C into a single manuscript. Given the minimal additional methods and results, and the major use of results from Parts A and B in the Part C discussion, it seems like the discussion in Part C could be condensed, without losing its purpose, and combined with Part B. Given I am not a reviewer on Part B, I will suggest the manuscript be reconsidered after major revisions. However, I will note that as a whole, Parts A, B, and C are a tremendous advance for our understanding of debris-covered glaciers, especially in Alaska. Therefore, if the editor believes Part C is warranted to provide sufficient space for the authors to discuss their two previous studies, then I would be supportive of accepting this manuscript subject to minor revisions.

Please find specific comments below.

Main Comments

Surface processes description: I disagree with the semantics used to describe surface processes as a separate term not explicitly referenced in the continuity equation, since they are explicitly in the continuity equation as the specific ablation. This description suggests that there is another term that needs to be accounted for. What the authors are trying to state is that surface processes are important since they control the distribution of ice cliffs, lakes, and streams, which feedback into the specific ablation and the ice dynamics. However, this feedback is nothing new and has already been described in L41-44. Furthermore, I would argue that "debris cover" should be included as a "surface process" because it differs from the typical clean ice and by itself would impact these relative feedbacks. I would recommend that the authors simply state that the specific ablation for debris-covered glaciers is affected by the distribution of debris thickness, ice cliffs, lakes, and streams, which will control the melt rate and feedback into the ice dynamics.

Accounting for streams that undercut cliffs: can the authors comment on how they handled mapping streams that are undercutting ice cliffs? Given the area of thick debris is more stagnant, this area has less ice cliffs. The ice cliffs that do exist are undercutting thicker debris which depending on the slope, may cause the ice cliff to be covered in a layer of debris (whether this suppresses or enhances melt is unknown), which is shown in Figure 8a and 9c. The key is that this region likely has thicker debris and fewer ice cliffs. The thicker debris means there is likely less backwasting at the top of the cliff compared to cliffs further upglacier that have thinner debris. This means that the cliffs may be able to survive longer. If the cliffs can survive longer, then they may be prone to have more streams that are undercut. I assume (the authors may confirm or deny) that these cliffs are unable to be mapped from high-resolution optical images. This could provide another explanation for the drop in the number of streams in the area of thick debris.

Specific Comments
*Italics* indicate suggested grammatical changes

L15 – "enhancing" the mass balance does not make sense. Consider changing mass balance to mass loss or enhanced to something like affected.

Abstract – a four paragraph abstract seems unnecessary. Consider condensing to one to two paragraphs.

L24 – "melt gradient" should be "melt rate gradients" to be consistent with the text.

L24-27 – the abstract should clearly reflect the main findings in the conclusion. I assume that the "high" in "high melt, melt gradients, and ice dynamics" means that all three of those elements are "high"? This is not particularly clear. Furthermore, what is a "high melt gradient" or "high ice dynamics"? Consider rephrasing these sentences, *making them more descriptive and easier to understand*. In its present form both the upper-limb and lower-limb have a high ice cliff and stream occurrence, which is inconsistent with the text. The conclusion states these feedbacks well. The abstract should do the same.

L28 – can you just state "The zone of maximum thinning occurs…" since the boundary between these two process domains is not well-defined anyways?

L34 – "insulates" surface melt does not make sense. Consider "insulates *the glacier* and strong reduces melt".

L44 – I would strongly encourage only using acronyms when they are absolutely necessary and common. I would recommend removing the acronym ZMT throughout the text to make it more readable for a broader audience.

L44 - Is Figure 1C a result of the present study or a result of Part B? If it is Part B, then it should be cited. If it's a result of this study, then the zone of maximum thinning should not be presented in the introduction.

Figure 1 – "with the opposite *sign* in the same pixel". State in the caption that the zone of maximum thinning is referenced by the double arrow. You can delete the ZMT as this is simply confusing in its present form and will be clear from the text. What does "Swatch profiles presented lower are 1000 m wide" mean? Where are these profiles? They do not appear to be shown in the figure. Also, the dH (dt)$_{-1}$ label looks very out of place. Consider positioning above the legend.

L45 – stating surface melt and ice dynamics are fundamental to thinning is repetitive of the prior paragraph and can be deleted.

L59 – somewhere in the introduction, whether this be the first sentence that uses "thick debris", or elsewhere, please define what is meant by "thick" debris (> 0.5 m? > 0.2m? >0.02 m?).

L66 and elsewhere – when referring to elevation make sure to be consistent. I would also recommend using "m a.s.l.".

L94 – what does "New analyses were required to estimate the annual velocity pattern" mean? Is this referring to Armstrong et al. (2016) and Armstrong et al. (2017)? Or the velocity maps produced in this study, which clearly was a new analysis?

L96 – based on what observation? This is really an assumption and should be stated as such.

L100 – define *w* in the text.

L110 – were the ice thickness "derived" or simply was ice thickness estimated by Huss and Farinotti (2012)?

L111 – Is this estimate of emergence rates assuming a uniform bed a second estimate of emergence rates? Or is this simply another assumption behind the emergence rate calculations? What does a uniform bed under the glacier fixed at the terminus mean?

Figure 4 is referenced before Figure 2 and 3. These should be placed in the order in which they are mentioned in the text.

Figure 2, Figure 5, and elsewhere – melt rate should always be positive. If the values are reported as negative then this should be the mass balance or surface lowering rate.

Figure 5 – why are the values placed on the right y-axis? This implies a secondary axis, but the only plot that has a true secondary axis is g. Change the labels to the left axis so that this plot is easier to read.  Unclear what "swatch profile" refers to. The description of the flat bed case in this caption should be moved to the text (L111). Change the following: Where surface velocities and emergence *rates are* low. I suggest explicitly pointing out the topographic bulge in panel e, so that this is clear for readers. Figure 5g - Is it necessary to abbreviate length to save two letters? This seems unnecessary. Also, confusing that the lakes are in a legend while the ice cliffs and streams are not. At a minimum the ice cliffs should be added to the legend, so that it is clear that they refer to the fractional area as well.

L125 – consider stating that the surface velocities decrease downglacier to near stagnation.

L129 – the range of emergence rates for both cases should be specified in the results.

L170 – "In the ablation zone" should be a new sentence.

L171 – rephrase this to be clearer. The key point here, which is explained well below, is that the feedback between the debris thickness controlling the melt rate, which affects the ice dynamics, which feedbacks to control the debris thickness.

L177 – close the parentheses.

L179 – should be a comma before "ice flow should also be high" and the same for the next sentence.

L182 – "melt rates are high, *and* surface slope…"

L187 – consider deleting the ":" and replacing with "as" or "since" to make it more readable.

L209 – this appears to be a universal statement. Is this meant for all debris-covered glaciers? Alaskan debris-covered glaciers? Are the authors confident with the 20 cm characterization despite the fact that they state the cutoff for these two process domains could be anywhere in the 10-20 cm range (L149)? A better preface could be that this mechanism is expected to occur on other debris covered glaciers where the debris transitions between the two process domains. Given the theory behind the discussion, this would seem to be more universal.

L216 – delete the comma.

L229 – "potentially *lead* to …"

L251 – Process links? Or Processes linked?

Figure 10 – Cause Ice Dynamics and Effect Debris have the same for the upper and lower limb. The text should be centered like the ice cliffs, lakes, etc. below it. Delete second "that" in caption.

L304 – should this be "debris thickness"?

---

## Referee Comment (RC2) · Anonymous Referee #2 · 18 Oct 2019

Review of Anderson et al., Part C, The Cryosphere, October 2019

In this third paper, Anderson et al. gathered ice velocity data and combine them with rough estimate of the ice thickness to infer ice fluxes and emergence velocities. They also derive the pattern of surface water streams on the glacier and their sinuosity. All the data collected in the three papers are then analysed to discuss feedbacks between ice dynamics and surface melt pattern and how they can explain the evolution of a debris covered glacier tongue.

General comments for the three papers (mostly similar to my review of part B).

1/ I am (really!) not convinced by the need to split this study into three parts. It implies

lot of repetitions and also mean that the reader as to refer to other parts of the article which is not convenient. Some data are plot several times in the three article (debris thickness, dh/dt for 1957-2009 etc. . .) I think the authors missed here an opportunity to put everything together. It would also help to convey more directly and simply the message. This is an exhausted reader (or reviewer) that finally reaches part C, a paper where very few news results are presented (just velocity data taken elsewhere and a map of the steam network that could have been presented at the time as the lake inventory). I found the discussion confusing and I must admit I did not understand the feedbacks at play. I also did not end up with a clear take home message.

2/ One strong limitation (that needs to be emphasized more) is that field measurements over a short period of time in July 2011 are used to interpret a map of elevation change over a multidecadal time period. Authors need to recall to their readers that their results apply to a short period of time. The whole discussion would have been much more meaningful if the elevation changes were also measured for the same time period where surface melt features are studied. Then, authors could have attempted to verify closure of the mass budget (continuity equation) between flux gates separating different parts of the glacier. It would have been a convincing verification of their surface melt estimates, involving some spatial extrapolation.

General comments for part C.

3/ I found a lot of speculation in the discussion. Just an example: that surface flow field has become more "S-shaped" through time. Authors do not present any velocity observation that can back up this. It seems to be just a good guess.

4/ A said above, the whole discussion is based on a zonation (the ZMT = zone of maximum thinning) of the glacier tongue from the long term dh/dt, over 5 decades. But to what extent this dh/dt rate is representative of the 2 month changes of the glacier? This is never addressed and it severely undermines the conclusions.

Specific comments.

Abstract does not really read like an abstract. More like an introduction. Authors should aim at ~250 words to keep it concise and to the point. There is no implication or general statements at the end.

L44. It was not demonstrated in part B that "ice dynamics control the location of the ZMT". This assertion comes from nowhere.

L77. "significant" is not quantitative. Can a percentage or a range of percentage be provided?

L88. "based off of" (?)

L110. How uncertain is this ice thickness data? Did this paper (or later studies by D. Farinotti) provide constrain on the (likely) large uncertainties for a single profile on glacier which is thinning rapidly. (when I see the nearly 0 emergence velocity in Figure 5 and the difference to the "flat bed" I think these uncertainties need to be discussed)

L114. At this stage in the paper, the reader wonders why streams need to be mapped. And why this is done in this third paper? Should ideally be grouped with lake mapping.

L115. Date of the image? digitization made for the entire glacier? Or the debris covered part only?

L120 the very limited amount of new result in this part C reinforces my opinion that this paper could be merged with other parts.

L125 can the authors confirm that this systematic offsets were not corrected ? and thus may result in biased emergence velocity? This is a significant proportion of the total velocity.

L129. I do not think these two cases of bed were described earlier in the text. Why the need for the Flat bed?

L155. The fact that debris thicken downglacier is probably repeated close to 10 times in the three papers (and also plot many times). This is irritating. It illustrates why the

artificial separation in three papers does not work.

Whole Section 4.1.2. I am not sure I get the point here and I do not really understand what is the actual finding: thick debris are found on almost all stagnating glacier tongues where melt rates are low, emergency velocity and dh/dt also. There is nothing really new here. Also I do not understand why the authors consider a steady state to interpret the evolution a glacier that is actually far from equilibrium. How debris are distributed nowaday is probably inherited from decades of imbalance.

L202-210. I find this part of the text poorly connected to the data/results obtained. Such a discussion would be relevant for a study examining time series of images and able to observe those debris mass wasting events related to the heterogeneity of the melt rate. Right now, no data in the study allow elaborating or confirming such a theory as a one-shot debris thickness and cliff distribution map was produced.

L310. Glacier

L311. "an ice cliff-glacier thinning feedback is evident on Kennicott Glacier". This was not evident at all for me, I do not think it was demonstrated or I did not get it.

Figures 1, 2 are good examples of redundant figures, already shown almost identically in part A and B.

Figure 3. Do the authors have evidences of reduced ice fluxes with time? This is probably an important part of the story, it is indicated on this figure but not really in the paper. Are the surface velocities changing with time? Or only the reduction in ice fluxes is due to surface lowering? These changes ice fluxes are probably key to understand the present-day distribution of dh/dt and debris on the tongue.

Figure 5d. The difference between "Flat and Variable" bed needs to be discussed more. It is worrisome that the "Flat Bed" curve show nearly 0 emergence velocity in region of high melt, in the active ice zone.

---

## Referee Comment (RC3) · Anonymous Referee #3 · 28 Oct 2019

This manuscript investigates the relation between the pattern of long-term thing of a debris covered glacier tongue in Alaska and the debris cover, surface features (cliffs, channels, ponds), flow dynamics. From this, it convincingly identifies and discusses in detail the emerging feedback between the related processes and quantities and thereby contributes to the very relevant and important discussion of what the role of glacier dynamics and surface features for the thinning (surprisingly high) of debris covered glaciers are. Some new results are also presented in this paper (channel mapping, sinuosity…) but the strength of this paper is the very systematic analysis and discussion of the different terms of the continuity equation that determines the thinning of a glacier (see fig 5). The undertaken bulking of the different quantities into few

zones (upper/lower limb ZWT,. . .) in the discussion of the results helps thereby to get a clearer picture and to identify the most dominant quantities and feedbacks. There are a few earlier papers available that tried to address the issue of anomalous thinning of debris covered glaciers but with a different approach (maybe that should be referenced better) and i think the identified importance of the reduced ice-emergence (reduced dynamic replacement) is very well supported by the presented data and analysis. Thus, overall this study presents a very interesting and important advance in understanding the dynamics of debris covered glaciers, a topic of high relevance in times of global warming, and hence this manuscript a very valuable contribution at TC. There are a few comments or and issues I have with this paper, but they are mostly rather minor (see list below) but would hopefully further improve an already very interesting, good quality and exiting paper. The figure and visualization are in general effective and the paper is well written.

More general comments 1. This paper is the last (part c) of a series of 3 papers, and one could always ask if not all parts should have been integrated into one paper. I admit that some repetition (in the description of some the data sets for example) is unavoidable, but for me part C works very well as a stand-alone paper and has a very clear own focus on the dynamic feedbacks and interactions and more than enough conclusive results for a stand-alone paper. I have to note that I only briefly looked into the other two papers (part A and B) but it was clear that there the main aim and focus of the otehr parts (A and B) were substantially different and in my view justified as separate papers. Moreover I believe that the main messages and findings of the three papers come in separate papers probably better across than in one huge one.

2. Abstract focus: somewhat related to the 3-part paper thing, when reading the abstract I got the impression that the main focus of the paper on the feedbacks and inter-relations between processes to explain the thinning pattern is rather thinly represented within the abstract (last 3 lines, and little on feedbacks but rather on correlations) and the results used in this papers but from part A and B get too much space in the abstract. A better balance and more focus on the feedbacks and findings of THIS part C would be useful.

3. Difference in time periods of datasets: One potential criticism of the analysis and conclusions one could have is that the thinning-data represents an average over several decades whereas the velocities and surface features, debris extent etc are the 'now' situation. I myself do not really think this is really a big issue but some more explanation and justification for this maybe useful.

4. Literature: With regard to influence/link of ice dynamics to thinning, debris cover and ice cliffs (e.g. explaining anomalous thinning) the paper by Banjeree (2017, TC), Rounce et al (2017), Ragettli (2016, TC) and potentially Moelg et al. (2019, TC) maybe useful to be considered.

Banerjee A. (2017): Thinning of debris-covered and debris-free glaciers in a warming climate. Brief communication. The Cryosphere, 11, 133-138, 2017 www.the-cryosphere.net/11/133/2017/ doi:10.5194/tc-11-133-2017 Rounce, D. R., King, O., McCarthy, M., Shean, D. E., & Salerno, F. (2018). Quantifying debris thickness of debris-covered glaciers in the Everest region of Nepal through inversion of a sub-debris melt model. Journal of Geophysical Research: Earth Surface,123, 1094-1115. https://doi.org/10.1029/2017JF004395 Ragettli, S., Bolch, T., & Pellicciotti, F. (2016). Heterogeneous glacier thinning patterns over the last 40 years in Langtang Himal, Nepal. The Cryosphere, 10(5), 2075-2097. https://doi.org/10.5194/tc-10-2075-2016 Moelg N., T. Bolch, A. Walter and A. Vieli (2019) Unravelling the evolution of Zmuttgletscher and its debris cover since the end of the Little Ice Age. The Cryosphere, 13, 1889-1909, https://doi.org/10.5194/tc-13-1889-2019

Minor/specific comments

p. 1 Line 15: the term 'melt hotspots' is here not really clear maybe specify a bit more what it is ('melt hotspots such as ice cliffs or channels')

p 1 lines 23-27: maybe make clearer what of these results from this part C paper and what is from earlier (or really focus on part C part).

P 1 line 24-25: high melt and HIGH melt gradients? here and also on next line it is not so clear to me what you mean by 'melt gradients' here, 'spatial gradients in melt' along flow, gradients in melt with regard to changing debris thickness. . .. be clearer.

P1 line 31: a brief explanation why ice cliffs are most abundant at the upglacier end maybe useful here (I think you have some idea about this or am I wrong?).

P 2 line 50: I think 'surface' uplift is here not quite correct, ice emergence is the relative movement to the surface or particle uplift against the surface, so maybe 'ice' uplift is more appropriate.

P 2 line 56-57: '. . .will facilitate the INTERPRETATION AND prediction of. . .'

P 2 line 62-63: importantly in part C you not just present data on ice dynamics and supraglacial streams but crucially in part C these data and all components of the mass conservation equation (thinning, flluy divergence. . .) are analysed for relation and feedbacks between them. Also say this here, as it is the backbone and most exiting part of this part C.

p. 3 lines 73-79: is this paragraph on the water pressure variations and sliding really needed? Maybe just summarize it in one sentence in the section 2.1 or 3.1 on the velocity data.

p 4 line 116: what is grid size chosen?

p. 5 lines 143-153: maybe this paragraph (together with next paragraph) can be shortened a little bit as already presented in part A and B.

p. 5 line 165: be clearer here on: 'THE ALONG FLOW/PROFILE PATTERN OF annual surface velocities. . ..

p. 6 line 171: this link between debris thickness and flow dynamics is a consequence

of the continuity equation, so maybe be more explicit on this. '. . .controls the melt rate and which a consequence of mass continuity is linked to ice dynamics.

p. 6 line 174-176: this complementation of debris thickness melt rates and surface velocity probably is meant in a steady state sense, otherwise the dhdebris/dt should also be mentioned (maybe clarify). Further to this sentence, with patterns I assume SPATIAL (along flow) PATTERNS are meant?

p. 6 line 183: again, '. . .SPATIAL /ALONG FLOW gradients in melt are low. . .'

p. 6 lines 186-191: good point!

p. 6 line 197-198: maybe explain why streams disappear in lower limb, is it because the drain through moulins in transition, but why are moulins there, connection to strain rates (longitudinal stretching? Not so clear in Fig. 5. )

p. 7 line 234: should it not be 'The lowest 4 km of the glacier ARE. . ..' (it is 4). And again why is this disconnection there, because water is drained though moulins to bed. . ..

p. 8 line 255. Maybe refer to Fig. 5b+f after (ZMT). Further with 'changes in ice flow' you probably mean along flow changes in ice flow, or more specifically the flux divergence or emergence rate.

p. 8 line 251: 'Process links . . .'

p. 9 line 273: not sure why you are so vague in your statement herewith 'may' be related. Why not be a bit more direct and say 'seems related to. . .'. Further do you mean to the 'SPATIAL/ALONG FLOW pattern'?

p. 9 line 290: 'increased ice strain', I struggle to see longitudinal extension (strain)here at the transition from the upper to the lower limb, the velocities clearly decrease down glacier there, so it would rather mean 'compression' or do I get something wrong here?

Fig. 1: the label dH/dt in the figure is rather confusing to read, make sure all is on one line e.g. dH*dtˆ(-1). Caption line 6: do you mean opposite 'sign' rather than 'sight'?

Caption line 7: 'The black line shows the profile used ....'

Fig. 2: I know that the exact threshold between upper and lower limb is not crucial but in the text a rough transition between 10cm and 20cm is given why not indicate this in the figure maybe as a grey shaded bar in the background.

Fig. 3: caption line one I would add at end of first sentence '... along the profile indicated in fig. 1c.'

Fig. 4: a detail but the map could do with a scale. More importantly, where in the figure/map is the box indicating the extent of Figure 11? I can simply not find it.

Fig. 5: in sub-fig (f) and in caption line 8, strictly speaking the label should be 'elevation change rate' as the sign is already negative and a lowering rate that is negative would then mean thickening again. Caption line 1: again make clear that the show data are '... for the swath along the profile indicated in fig. 1c.'

Fig. 7: here an elevation threshold/bands are used to summarize/group the sinuosity data, but am I right that these are both above the ZMT and in the upper limb of the oestroem curve. This maybe useful to be explained in the caption.

Fig. 10. I found this figure rather difficult to read, there is a lot of information and detail and I initially expected from this schematic to better get the big picture. Maybe I just expected the wrong thing and the colors (blue or red) were not so clear to me and I wondered if it really helped me a lot. If I see it as complete documentation of all different relations and feed backs it is maybe fine, but then maybe it should be phrased as such. More importantly, in the caption the colors red and blue refer to positive effects or negative effects but it is not so obvious to me what you mean by positive and negative. Does this refer to positive and negative FEEDBACKS (self enhancing/reducing) or positive/negative from a glacier health (negative mass loss, reduced speed,....). should be clarified.

---

## Referee Comment (RC4) · Martin Kirkbride (Referee) · 4 Nov 2019

This interesting and topical paper synthesizes a range of glaciological data to improve understanding of the process feedbacks between glacier flow, melt distribution under debris cover, and thinning, at a large compound Alaskan glacier. The ambition of the paper is welcome: there is an increasing output of papers dealing with one or two aspects of debris-covered glacier (DCG) monitoring and evolution, many based on state-of-the-art data gathering, but few attempts have hitherto been made to understand interactions at appropriate timescales, and to come up with integrative explanatory models. The paper bases its approach on mass continuity and the debris-thickness/melt

relationship (the Ostrem Curve).

My general comment is that this is a rigorous and well-argued study which shows some interesting results, different from other papers I am familiar with. The core finding is that the interaction of ice flow, debris emergence, melt and thinning have produced a subtle "bulge" several kilometres above the terminus, marking the transition from active ice flow and debris emergence upstream to relatively stagnant, heavily debris-covered ice downstream. My surprise is that the active/stagnant transition is manifest as a convexity in the long profile, rather than a concavity as described in DCGs elsewhere. While Figs 2 and 5 are vertically exaggerated to show this subtle topographic evolution (as they must be), it is convincingly demonstrated. The transition corresponds to the kink in the downward limb of the Ostrem Curve at which the rate of sub-debris melting becomes less sensitive to debris thickness.

The paper raises some interesting questions, but also contains some inferences of cause-effect which are less well substantiated than others. There is perhaps a tendency in places to make easy inferences of causation based on only the available data, when other variables have not been considered. (This is not to denigrate the high-quality datasets presented). As such, I don't think it provides definitive answers to the problem of quantifying the feedbacks in these complex systems, but it does point to a way forwards.

Another issue (also in no way a criticism here) is that the literature presents the "debris-covered glacier" as if it is a single class of glacier: this is not the case. DCGs take many forms and origins, and are unlikely to have a single unifying model of behaviour and evolution. This study of Kennicott Glacier is of a very large compound valley glacier terminating in a proglacial lake, whose debris cover is fed by coalescing medial moraines. We might not expect models from this glacier to apply easily to (for example) smaller moraine-dammed DCGs whose flow is obstructed towards the terminus, or single-basin glaciers with transverse foliation. Perhaps some acknowledgement of this diversity would be appropriate.
It isn't clear from this paper (Part C of three) what the ice thickness distribution is, but this information would be useful. This is because, while velocity evolution is a key variable, the causes of velocity change and its distribution on the long profile are not covered, yet this information is essential for understanding the dynamic evolution of the glacier. I would like to see some consideration of the effects of both thinning rates and surface gradient changes on the driving stresses, to explore why the observed pattern of stagnation has developed: it implies a collapse in the driving stress from the terminus upstream, which in turn must be some combination of reduced ice thickness and slope. It is noteworthy (though largely unrecognised generally) that very thick, very gentle glaciers such as DCG tongues are sensitive to small changes in slope, at least as much as in thickness. So there is scope for a fuller explanation than is given in the manuscript.

I have some minor line-by-line comments to improve the presentation, and to correct minor editorial mistakes (attached).

Please also note the supplement to this comment:
https://www.the-cryosphere-discuss.net/tc-2019-178/tc-2019-178-RC4-supplement.pdf

**Supplement:**

**Anderson et al: "Debris cover and the thinning of Kennicott Glacier (Part C)…."**

**Detailed minor comments**

Line
How can mass balance be "enhanced": rephrase.
Need to define upper limb and lower limb of Østrem's curve, because what is referred to here are really segments of the same limb (debris thicker than effective thickness). Don't hyphenate "upper limb" or "lower limb".
Suggest "in spite of" instead of "as well as"?
"may…control": it clearly does!
36-7    Why is the term "melt hotspots" in italics? Unnecessary.
Although the term "debris-cover anomaly" has gained currency since 2015, there is often a careless use of terminology in this context, where glacier thinning and melting are used synonymously. The "anomaly" (if one exists" is in the thinning rates, not the sub-debris melt rates. Make this clear.
"causes", not "cause" (process is singular).
62-3    One cannot estimate a supraglacial stream. Rephrase.
"south-facing"
Suggest "more cliffs per unit area".
Add comma after "significantly".
Add hyphen after "column".
Remove hyphen after "corrected".
I take issue with the use of "bi-modal" here, because a bimodal distribution has two modes (peaks). Here, the term is used to indicate an absence of streams on thickly debris-covered ice: this isn't "bimodal", rather it's a threshold control.
147-152 Remove hyphens in "upper limb" and "lower limb". See comment re. line 24 about clarity of what these terms mean.
Remove italics: unnecessary.
See l. 147
Why use the term "attractor state" here? You imply the glacier is attracted to an equilibrium state of mass balance, but there's no reason for this to be more likely than any other mass balance state because mass balance is not controlled by internal system dynamics.
Commas after "are high" and "are low".
Re. chicken-egg quandary: this disappears if a longer-term view is taken, in which velocity is the ultimate control, because the glacier must slow down to allow debris cover to accumulate ("ablation-dominant" conditions of Kirkbride (2000 IAHS))". So the question becomes what causes change to the longitudinal velocity profile of the glacier over time, where does velocity reduce earliest on this profile, and why? (See my general comments).
See l. 135
See l. 135
et seq. It's really no surprise that streams are more abundant on steeper gradients, and lakes on gentle gradients, since water flows downhill. What point is being made here?
I'm perplexed by the conclusion that ice cliff abundance is related to basal sliding rate. I simply don't see a direct connection here, and wonder whether you are taking spatial associations too far down the line of causal relationships. If the connection is indirect, it needs to explained clearly and in full.
I don't understand how stream undercutting od ice walls increases debris thickness at the base of the ice slope. This implies that the ice slopes must decline in angle, for which no evidence is given: parallel retreat will give the same thickness at the base as at the top. (More likely, fluvial removal of debris occurs, so an apparent thickening as seen in Fig 9 may be debris brought to the site from upstream). Suggest omitting these two sentences.

I disagree that the lower glacier is "hydrologically disconnected". Supraglacial drainage becomes englacial (and subglacial?) which isn't the same as being disconnected (see Fyffe et al 2019 *J Hydrol* 570, 584-597).

"Ice cliffs are … more likely to be buried". Buried how? This assumes a process of disappearance which isn't explained. I agree with the general point about their removal, but the process needs careful explanation.

Debris cover and surface drainage basin relationships are shown nicely in Catriona Fyffe's recent paper (see l. 234 comment).

The effect of this slope reduction is probably a key observation, because on thick, gentle glaciers the driving stress can be at least as sensitive to small changes in slope as to ice thinning. It would be interesting to see how this slope reduction plays out with changes to the basal stress profile over time, which may show something useful re. velocity.

See l. 242: on steep, thin glaciers, thickness change is the main control: on gentle, thick glaciers, slope is more important. Perhaps refine this sentence in the context that DCGs are characteristically thick and gentle.

Replace "pattern of debris" with "distribution of debris thickness": be specific.

"… this pattern over time"

Desperately needs a comma after " thinned" , otherwise the sentence makes no sense.

"have", not "has". The stated change in the surface flow field is not supported by any evidence. Either omit this point, or provide evidence for it. If true (which I'm sure it is), clean ice would be redistributed as well as debris-covered ice, so is it an explanation at all?

See l.147

Spelling "Glaicer"

"DS" is acknowleged here, but isn't a named author of the paper.

Captions

Fig 1     Panel (a) doesn't show the location of Panel (b).

Fig 2.     I would go further in saying the elbow of the curve lies between 12 and 14 cm. Could you fit best-fit lines to each segment iteratively to find the location of the angle? Also, highlight the bare ice point more clearly. Which altitude does this point originate from? (it can't be a unique point).

Fig 5.     The key figure in the paper, and really interesting to absorb. One query is why in (e) the elevation difference decreases below c. 3km above the terminus, but in (f) the surface lowering rate increases over the same distance? This seems inconsistent.

Fig 7.     While interesting in its own right, I'm sure what data on stream sinuousity contributes to the overall interpretations and conclusions. This figure and the accompanying text could be omitted, unless a stronger case is made for its inclusion.

---

## Author Comment (AC1) · 15 Feb 2020

Review of "Debris cover and the thinning of Kennicott Glacier, Alaska, Part C: feedbacks between melt, ice dynamics, and surface processes" by Anderson et al.

Thank you kindly for taking the time to review our manuscripts.

This study is the third part of three publications that investigate debris cover on Kennicott Glacier in Alaska. The focus of this study is on the feedbacks between the melt, ice dynamics, and surface processes for the debris-covered portion of Kennicott Glacier. After reading all three parts, the introduction feels repetitive of Parts A and B (I recognize this is unavoidable).

We will work to make the introductions distinct between the three contributions.

Unfortunately, the methods and results in this section are highly underwhelming. The new results in this part are surface velocities, emergence velocities based on those surface velocities and ice thickness, and manual delineation of streams for a single WorldView scene. These are all quite straightforward analyses that are fairly easy to perform.

We agree that these are easy analyses, but the strength of this work is in bringing the pieces together and providing a number of new, important process observations. We do propose a number of improvements to enhance this third part of the paper series.

Where this paper excels is in its discussion, which is grounded in the observations and results from Parts A and B with a little support from results and theory in Part C.

We want to also emphasize the number of new process descriptions we add here. While many of these describe processes acting on the glacier, quantifying these behaviors in the future studies may be an important future direction of work on debris-covered glaciers.

While the discussion does not provide any conclusions that are necessarily groundbreaking  (the relationship between debris thickness, surface velocities, and surface processes have been detailed for debris-covered glaciers in other parts of the world in other studies, which this manuscript references) … attempts to make universal statements concerning all debris-covered glaciers at times.

We want to emphasize how much new material is actually in this manuscript. These are observations that have never been put together on a single glacier, with the continuity equation kept in mind. Other studies have discussed some feedbacks between some surface processes but here we also provide new process descriptions.

From reviewer 3:

"for me part C works very well as a stand-alone paper and has a very clear own focus on the dynamic feedbacks and interactions and more than enough conclusive results for a stand-alone paper. "

… attempts to make universal statements concerning all debris-covered glaciers at times.

While we appreciate that there is a lot of diversity of debris-covered glaciers we also feel that there are fundamental processes and physics that are acting on all debris-covered glaciers. We try to walk

a line between emphasizing these fundamental processes that may occur on other debris-covered glaciers. We will take more care though in the revision process to ensure that we don't overstep with our assertions.

The manuscript is well written, the figures support the text well, and there are sufficient references to the existing literature. Hence, my comments are fairly minor, but I admittedly have mixed feelings concerning the originality of the paper to stand on its own.

Based on

From reviewer 3:

"for me part C works very well as a stand-alone paper and has a very clear own focus on the dynamic feedbacks and interactions and more than enough conclusive results for a stand-alone paper. "

If Parts A and B were separate studies by other authors, then I would argue that the originality and methodology would be poor-fair; this paper would come across more as a review paper of how existing studies are connected and likely not warrant publication without major revisions. However, that is not the case, and instead this paper comes across as an extension of Parts A and B, and a place where everything can be discussed in a broader context.

What would truly elevate this paper to stand on its own, would be if the theoretical feedbacks were supported by model results. Given that Part B develops empirical equations for accounting for distributed melting due to debris, ice cliffs and backwasting, this would be a very logical next step. However, I recognize that more information is likely needed concerning ice cliff nucleation and debris redistribution to be able to model these various surface processes over long periods of time and at a high enough level to support the discussion.

We would love to model the feedbacks we describe here and that is the target of further research. Here though our aim is to provide process descriptions beyond the important feedbacks that have been explored extensively in these manuscripts as well as in others.

My recommendation would be to integrate Parts B and C into a single manuscript. Given the minimal additional methods and results, and the major use of results from Parts A and B in the Part C discussion, it seems like the discussion in Part C could be condensed, without losing its purpose, and combined with Part B. Given I am not a reviewer on Part B, I will suggest the manuscript be reconsidered after major revisions. However, I will note that as a whole, Parts A, B, and C are a tremendous advance for our understanding of debris-covered glaciers, especially in Alaska. Therefore, if the editor believes Part C is warranted to provide sufficient space for the authors to discuss their two previous studies, then I would be supportive of accepting this manuscript subject to minor revisions.

Thank you kindly for your time and effort reviewing these manuscripts. We appreciate it and hope to return the favor soon. While we understand the desire for consolidation we also feel that that this reviewer actually missed the new feedback we reveal on Kennicott Glacier.

We also want to highlight that this is the first study we know of that rather clearly links ice dynamics to ice cliff distribution.

Please find specific comments below.

Main Comments
Surface processes description: I disagree with the semantics used to describe surface processes as a separate term not explicitly referenced in the continuity equation, since they are explicitly in the continuity equation as the specific ablation. This description suggests that there is another term that needs to be accounted for. What the authors are trying to state is that surface processes are important since they control the distribution of ice cliffs, lakes, and streams, which feedback into the specific ablation and the ice dynamics. However, this feedback is nothing new and has already been described in L41-44.

The feedback we are highlighting is actually different than what the reviewer has just quoted and is outlined in Vincent et al., 2016; and Brun et al., 2018. This may be because we did not clearly describe the feedback.

On Kennicott Glacier we find that we find that ice dynamics appears to correlate with ice cliff dinsity. The process links we describe must not be clear enough though in the manuscript as it stands now. We will clarify this in our writing. The high strain rate at the upper part of the (zone of maximum thinning) *ZMT* correlates with high ice cliff density. Low strain rate we see low ice cliff density. We provide a physical mechanism where high strain rates can lead to increased ice cliff density. Increased ice cliff density leads to increased melt rates, which then contribute to increased glacier thinning locally at the upper end of the *ZMT*. Active ice dynamics and increased emergence rates which tend to locally thicken the glacier are compensated with increased ice cliff coverage which tends to thin the glacier.

This is an absolutely new feedback that we have identified. Clearly we need to describe this effect more clearly.

Furthermore, I would argue that "debris cover" should be included as a "surface process" because it differs from the typical clean ice and by itself would impact these relative feedbacks. I would recommend that the authors simply state that the specific ablation for debris-covered glaciers is affected by the distribution of debris thickness, ice cliffs, lakes, and streams, which will control the melt rate and feedback into the ice dynamics.

We appreciate this perspective. But we are taking a view that is more from geomorphology (earth surface processes in general). From a landscape evolution perspective the erosion of the earth's surface is the glaciological equivalent of melt. The actors causing the erosion of the earth's surface would be rivers, mass wasting, and hillslope processes. But on debris-covered glaciers melt is the result of heat from the atmosphere, and solar radiation and also other features, like ice cliffs, streams, and ponds. We will think about this differentiation going into revisions, though.

Accounting for streams that undercut cliffs: can the authors comment on how they handled mapping streams that are undercutting ice cliffs?

Thank you for highlighting this, we can be more clear. We will show WV photos from the glacier surface that show the processes of extrapolation under ice cliffs, this can be included in the supplemental. We have field photos to show the extreme sinuosity of many of these streams. That guided our digitization.

Given the area of thick debris is more stagnant, this area has less ice cliffs. The ice cliffs that do exist are undercutting thicker debris which depending on the slope, may cause the ice cliff to be covered in a layer of debris (whether this suppresses or enhances melt is unknown), which is shown in Figure 8a and 9c.

Yes we agree with these statements.

The key is that this region likely has thicker debris and fewer ice cliffs. The thicker debris means there is likely less backwasting at the top of the cliff compared to cliffs further upglacier that have thinner debris. This means that the cliffs may be able to survive longer.

Thanks for highlighting this processes we will try to work it in as a possibility.

If the cliffs can survive longer, then they may be prone to have more steams that are undercut. I assume (the authors may confirm or deny) that these cliffs are unable to be mapped from high-resolution optical images. This could provide another explanation for the drop in the number of streams in the area of thick debris.

We walked over most of this lower tongue and there are very few streams present at the base of these cliffs where debris is thick. We will provide photo evidence to support this.

There just aren't streams in this region with thick debris and the ice cliffs are often in closed depressions and have small drainage basins, which we will more clearly highlight with analysis of a high resolution surface DEM.

Specific Comments
Italics indicate suggested grammatical changes

L15 – "enhancing" the mass balance does not make sense. Consider changing mass balance to mass loss or enhanced to something like affected.

We will change this term.

Abstract – a four paragraph abstract seems unnecessary. Consider condensing to one to two paragraphs.

Yes, we can do this.

L24 – "melt gradient" should be "melt rate gradients" to be consistent with the text.

L24-27 – the abstract should clearly reflect the main findings in the conclusion. I assume that the "high" in "high melt, melt gradients, and ice dynamics" means that all three of those elements are "high"? This is not particularly clear. Furthermore, what is a "high melt gradient" or "high ice dynamics"?

We will make this more clear.

Consider rephrasing these sentences, making them more descriptive and easier to understand. In its present form both the upper-limb and lower-limb have a high ice cliff and stream occurrence, which is inconsistent with the text. The conclusion states these feedbacks well. The abstract should do the same.

We will clarify as suggested.

L28 – can you just state "The zone of maximum thinning occurs..." since the boundary between these two process domains is not well-defined anyways?

We can more definitively define the boundary between these zones.

L34 – "insulates" surface melt does not make sense. Consider "insulates the glacier and strong reduces melt".

We can clarify this.

L44 – I would strongly encourage only using acronyms when they are absolutely necessary and common. I would recommend removing the acronym ZMT throughout the text to make it more readable for a broader audience.

We will consider removing the acronym.

L44 - Is Figure 1C a result of the present study or a result of Part B? If it is Part B, then it should be cited. If it's a result of this study, then the zone of maximum thinning should not be presented in the introduction.

We will cite Part B.

Figure 1 – "with the opposite sign in the same pixel". State in the caption that the zone of maximum thinning is referenced by the double arrow. You can delete the ZMT as this is simply confusing in its present form and will be clear from the text.

We will do this.

What does "Swatch profiles presented lower are 1000 m wide" mean? Where are these profiles? They do not appear to be shown in the figure.

The swath profile is 500 m on either side of the line in Figure 1c.

Also, the dH (dt) -1 label looks very out of place. Consider positioning above the legend.

We will move this.

L45 – stating surface melt and ice dynamics are fundamental to thinning is repetitive of the prior paragraph and can be deleted.

L59 – somewhere in the introduction, whether this be the first sentence that uses "thick debris", or elsewhere, please define what is meant by "thick" debris (> 0.5 m? > 0.2m? >0.02 m?).

We will define this.

L66 and elsewhere – when referring to elevation make sure to be consistent. I would also recommend using "m a.s.l.".

We will use this.

L94 – what does "New analyses were required to estimate the annual velocity pattern" mean? Is this referring to Armstrong et al. (2016) and Armstrong et al. (2017)? Or the velocity maps produced in this study, which clearly was a new analysis?

We refer to the mean annual velocity derived from the Part C study.

L96 – based on what observation? This is really an assumption and should be stated as such.

We will rephrase this but we aren't sure it is really an assumption as the equation is simply from fluid mechanics.

L100 – define w in the text.

We will do this.

L110 – were the ice thickness "derived" or simply was ice thickness estimated by Huss and Farinotti (2012)?

It is the ice thickness from Huss and Farinotti. We will clarify this.

L111 – Is this estimate of emergence rates assuming a uniform bed a second estimate of emergence rates? Or is this simply another assumption behind the emergence rate calculations? What does a uniform bed under the glacier fixed at the terminus mean?

We will clarify this in the text. For the uniform bed case, we assume that the bed of Kennicott Glacier is uniform at the elevation of the terminus mean. We include this just to show an end member case.

Figure 4 is referenced before Figure 2 and 3. These should be placed in the order in which they are mentioned in the text.

Figures 2 and 3 are references on line 62. Figure 4 is referenced on line 125. No change is needed.

Figure 2, Figure 5, and elsewhere – melt rate should always be positive. If the values are reported as negative then this should be the mass balance or surface lowering rate.

Thank you for pointing this out. We will correct this throughout the manuscripts.

Figure 5 – why are the values placed on the right y-axis? This implies a secondary axis, but the only plot that has a true secondary axis is g. Change the labels to the left axis so that this plot is easier to read.

Thanks for this comment. We put the tick labels on the right because most of the data is 'high' in the plot on the right and low on the left. So it isn't clear that putting the labels on the left will improve the readability of the plot.

Unclear what "swatch profile" refers to.

The center of the swath profile is shown in Figure 1C.

The description of the flat bed case in this caption should be moved to the text (L111). Change the following: Where surface velocities and emergence rates are low. I suggest explicitly pointing out the topographic bulge in panel e, so that this is clear for readers.

Yes we will make these changes.

Figure 5g - Is it necessary to abbreviate length to save two letters? This seems unnecessary.

Maybe not we will take a look, but we did this so the label was not fixed to three lines.

Also, confusing that the lakes are in a legend while the ice cliffs and streams are not. At a minimum the ice cliffs should be added to the legend, so that it is clear that they refer to the fractional area as well.

We will work on this, but there just isn't enough space in the panel, without reducing the legend text. We will try.

L125 – consider stating that the surface velocities decrease downglacier to near stagnation.

Yes, thank you.

L129 – the range of emergence rates for both cases should be specified in the results.

L170 – "In the ablation zone" should be a new sentence.

L171 – rephrase this to be clearer. The key point here, which is explained well below, is that the feedback between the debris thickness controlling the melt rate, which affects the ice dynamics, which feedbacks to control the debris thickness.

We will make sure this is clear.

L177 – close the parentheses.

L179 – should be a comma before "ice flow should also be high" and the same for the next sentence.

L182 – "melt rates are high, and surface slope..."

L187 – consider deleting the ":" and replacing with "as" or "since" to make it more readable.

L209 – this appears to be a universal statement. Is this meant for all debris-covered glaciers? Alaskan debris-covered glaciers? Are the authors confident with the 20 cm characterization despite the fact that they state the cutoff for these two process domains could be anywhere in the 10-20 cm range (L149)? A better preface could be that this mechanism is expected to occur on other debris covered glaciers where the debris transitions between the two process domains. Given the theory behind the discussion, this would seem to be more universal.

We were not clear enough in writing. What we mean is that ice cliffs are more likely to be buried the thicker the debris is around the ice cliff. The debris climbs up the ice cliff. We will provide videos to support his inference. We have two supplementary videos to support this.

L216 – delete the comma.

L229 – "potentially lead to ..."

L251 – Process links? Or Processes linked?

We will clarify this.

Figure 10 – Cause Ice Dynamics and Effect Debris have the same for the upper and lower limb. The text should be centered like the ice cliffs, lakes, etc. below it. Delete second "that" in caption.

We will correct this.
L304 – should this be "debris thickness"?

Thank you we will correct this.

**Part C: proposed changes**

We want to emphasize here that we do outline new feedbacks in this paper.

From Reviewer 3 from Part C:

"P 2 line 62-63: importantly in part C you not just present data on ice dynamics and supraglacial streams but crucially in part C these data and all components of the mass conservation equation (thinning, flux divergence. . .) are analysed for relation and feedbacks between them. Also say this here, as it is the backbone and most exiting part of this part C."

On Kennicott Glacier there is a strong correspondence between ice cliffs and active ice flow. While weak relationships have been suggested here on Kennicott the correlation is more clear than anywhere else.

The highest concentration of ice cliffs occurs at the upper end of the zone of maximum thinning. The high concentration of ice cliffs also corresponds to where we expect ice emergence rates to be high. These ice emergence rates uplift the glacier surface, working to counter glacier thinning. But ice dynamics, which produce this surface uplift also seems to produce more ice cliffs (see the physical descriptions within the main article). These ice cliffs counter the effect of surface uplift, they are essentially a negative feedback on the effect of ice dynamics.

In addition to this new feedback we also present a number of new hypotheses for the interaction of surface processes with melt and ice dynamics with a new, holistic perspective.

We feel that there is more than enough new material here for a stand alone paper, but in order to improve the manuscript we propose that we add these additional datasets/ideas to Part C:

- New annual surface velocities from 2000-2010
  - These velocities allow us to calculate changes in ice emergence rate and ice flux over the in situ measurement period
  -More detailed discussion of the reduction of ice emergence rate through time.

- Delineation of drainage basins on the glacier surface (new figure) to support the stream story already within the manuscript.

- Tie in a discussion about glacier surface topography. Ice cliff maximum heights (from in situ measurements), the number of individual ice cliffs with elevation band, and calculated glacier surface relief down glacier.

- New processes drawings to show the important new observations that we are highlighting in this paper. This will greatly improve the reader's ability to see the new process links we are describing.

- Additional photo evidence from the field outlining these new processes links. Many will go into the supplemental but they will support and clarify the process links we are highlighting.

- Description of a new ice cliff burial mechanism. Timelapse movies from the Kennicott and Ngozumpa glaciers (in the supplemental) showing a new mechanism for the burial of ice cliffs. The actual process is not yet described in detail in the text.

- A paragraph that is the same for each of the 3 parts that outlines how they build off of one another.

---

## Author Comment (AC2) · 15 Feb 2020

Thank you kindly for taking the time to review our manuscripts! We appreciate it.

Review of Anderson et al., Part C, The Cryosphere, October 2019

In this third paper, Anderson et al. gathered ice velocity data and combine them with
rough estimate of the ice thickness to infer ice fluxes and emergence velocities. They
also derive the pattern of surface water streams on the glacier and their sinuosity. All
the data collected in the three papers are then analysed to discuss feedbacks between
ice dynamics and surface melt pattern and how they can explain the evolution of a
debris covered glacier tongue.

Thank you kindly for taking the time to review this manuscript. We appreciate your comments and
thoughts.

General comments for the three papers (mostly similar to my review of part B).
1/ I am (really!) not convinced by the need to split this study into three parts. It implies lot of
repetitions and also mean that the reader as to refer to other parts of the article
which is not convenient. Some data are plot several times in the three article (debris
thickness, dh/dt for 1957-2009 etc. . .) I think the authors missed here an opportunity
to put everything together. It would also help to convey more directly and simply the message.

We appreciate your efforts in reviewing these manuscripts. But we feel that the reviewer does not
appreciate the insights provided by each of the three manuscripts that outlines different aspects of a
large debris-covered glacier from a region where there are no other studies on debris covered glacier
mass balance.

From reviewer 1:

"If Parts A and B were separate studies by other authors, then I would argue that the originality and
methodology would be poor-fair; this paper would come across more as a review paper of how
existing studies are connected and likely not warrant publication without major revisions. However,
that is not the case, and instead this paper comes across as an extension of Parts A and B, and a
place where everything can be discussed in a broader context."

From reviewer 3:

"for me part C works very well as a stand-alone paper and has a very
clear own focus on the dynamic feedbacks and interactions and more than enough
conclusive results for a stand-alone paper. "

From reviewer 4:

"My general comment is that this is a rigorous and well-argued study which shows some

interesting results, different from other papers I am familiar with."

This is an exhausted reader (or reviewer) that finally reaches part C, a paper where very few news results are presented (just velocity data taken elsewhere and a map of the steam network that could have been presented at the time as the lake inventory). I found the discussion confusing and I must admit I did not understand the feedbacks at play. I also did not end up with a clear take home message.

This is an ambition manuscript and the first manuscript to tie a diversity of measurements together with the continuity equation. This in and of itself is a very valuable contribution, especially from a glacier that shows clearly that these components co-vary.

The point is to connect the melt pattern, ice dynamics, and surface processes in a fashion that has not yet been done. We outline a new potential feedback between ice cliffs and thinning that has not yet been identified. We think that that there are quite a few take home messages from this manuscript. For example we show important feedbacks between streams and ice cliffs, new potential processes for the formation of ice cliffs, new links between ice dynamics and ice cliff distribution. These are all new, important contributions just from Part C.

If we were to combine all three parts most of these insights would be lost along the way. Furthermore there would be so many directions explored in a single manuscript that it would be even more overwhelming and less legible.

2/ One strong limitation (that needs to be emphasized more) is that field measure-
ments over a short period of time in July 2011 are used to interpret a map of elevation
change over a multidecadal time period. Authors need to recall to their readers that
their results apply to a short period of time. The whole discussion would have been
much more meaningful if the elevation changes were also measured for the same time
period where surface melt features are studied.

We have dh/dt data that spans the in situ measurement period and we will include them in Part B. The zone of maximum thinning is in the same location as the dh/dt map we show in Part B and C.

The extensive discussion at the end of Part B outlines how extreme the changes in ice cliff coverage (increase in ice cliff coverage of 70%) and debris thickness (90% reduction in mean debris thickness) needed to create maximum melt in the ZMT. We feel that this is a compelling argument. We are surprised that this argument was not clear.

Then, authors could have attempted to verify closure of the mass budget (continuity equation)
between flux gates separating different parts of the glacier. It would have been a convincing
verification of their surface melt estimates, involving some spatial extrapolation.

We realize that past work took the approach you outline here, but we want to emphasize that different ways of argument we use are also viable. The line of argument we use is an alternative, viable approach to flux gates and DEM differencing over the same time span.

General comments for part C.

3/ I found a lot of speculation in the discussion. Just an example: that surface flow
field has become more "S-shaped" through time. Authors do not present any velocity
observation that can back up this. It seems to be just a good guess.

We would rather posit that some of the paper is based off of field observations, while we do not model or provide quantitative constraints on the hypotheses/observations they are still new, important contributions.

The medial moraines tend to follow flow lines. But yes we are including new surface velocity estimates in the next version of the paper.

4/ A said above, the whole discussion is based on a zonation (the ZMT = zone of maximum thinning) of the glacier tongue from the long term dh/dt, over 5 decades. But to what extent this dh/dt rate is representative of the 2 month changes of the glacier? This is never addressed and it severely undermines the conclusions.

Thank you for bringing up this discrepancy. We addressed this discrepancy in length in Part B in the discussion by placing extreme bounds on our melt estimates. We are including a more recent dh/dt that spans the field measurement period (the ZMT is in basically the same location).

Specific comments.

Abstract does not really read like an abstract. More like an introduction. Authors should aim at ∼250 words to keep it concise and to the point. There is no implication or general statements at the end.

We will improve the abstract, but the holistic perspective we take on this work is new and we provide a way forward for the research of debris-covered glaciers. As debris-covered glaciers researchers it is an important next step for us to start looking at interactions between the components of the debris-covered glacier system.

L44. It was not demonstrated in part B that "ice dynamics control the location of the ZMT". This assertion comes from nowhere.

We will justify this more clearly. From a simple process of elimination from the continuity equation and our extreme uncertainty analysis in Part B we thought this was apparent.

L77. "significant" is not quantitative. Can a percentage or a range of percentage be provided?

Yes we will add the percentage here.

L88. "based off of" (?)

We will change this to 'derived from'

L110. How uncertain is this ice thickness data? Did this paper (or later studies by D. Farinotti) provide constrain on the (likely) large uncertainties for a single profile on glacier which is thinning rapidly.

Yes, We are happy to discuss those uncertainties. But it will not change the pattern of ice emergence rates that we estimate, unless the glacier is thicker below than ZMT than above it.

(when I see the nearly 0 emergence velocity in Figure 5 and the difference to the "flat bed" I think these uncertainties need to be discussed)

But ice emergence rates are derived from dQ/dx and Q = $\bar{u} * H$   If   $\bar{u}$   is very low then it doesn't really matter what *H* values are Q will still be low.

L114. At this stage in the paper, the reader wonders why streams need to be mapped.
And why this is done in this third paper? Should ideally be grouped with lake mapping.

We will make the justification for presenting streams in Part C in the intro. We will also add a back of the envelope estimate of the effect of streams on mass balance on Kennicott Glacier in Part B.

This is included in the third paper because streams up to this point have not been included as significant contributors to surface mass balance. How would we quantify their effect on surface mass balance? We intend to include a back-of-the-envelope calculation of the potential effect of streams on the Kennicott Glacier in Part B.

But we also prefer to have the stream digitization here in Part C where we discuss what we see as the primary role of streams: maintaining ice cliffs and focusing debris. And stream sinuosity plays a potentially important role in those effects.

L115. Date of the image? digitization made for the entire glacier? Or the debris covered part only?

We will clarify this.

L120 the very limited amount of new result in this part C reinforces my opinion that this paper could be merged with other parts.

We disagree here. Where has a similar study been published? It would be helpful if the reviewer provided evidence for this statement.

Velocity, melt and surface features have very clear patterns and some co-vary. This starkness of these patterns is worth in depth discussion because we are unaware of any other studies that so these relationships so clearly and consistently. Additionally we add a number of new process feedbacks that have not yet been detailed.

We provide a first holistic view of the debris-covered glacier system. While yes it has been shown that ice dynamics are important for debris-covered glacier thinning, in this manuscript we want to move beyond these items viewed as isolated entities but rather as pieces of a whole that interact with one another.

The stack of data (Fig. 5) is rather unique and not where we tie the different components together. The discussion about ice cliffs and streams and feedbacks is completely new.

Much of the new insight comes from field observations. But these are important observations that need to be published.

From reviewer 1:

"If Parts A and B were separate studies by other authors, then I would argue that the originality and methodology would be poor-fair; this paper would come across more as a review paper of how existing studies are connected and likely not warrant publication without major revisions. However,

that is not the case, and instead this paper comes across as an extension of Parts A and B, and a place where everything can be discussed in a broader context."

From reviewer 3:

"for me part C works very well as a stand-alone paper and has a very clear own focus on the dynamic feedbacks and interactions and more than enough conclusive results for a stand-alone paper. "

From reviewer 4:

"My general comment is that this is a rigorous and well-argued study which shows some interesting results, different from other papers I am familiar with."

L125 can the authors confirm that this systematic offsets were not corrected ? and thus may result in biased emergence velocity? This is a significant proportion of the total velocity.

Sure we can correct for this error but it is a systematic error. But it is not clear how a systematic offset would change the **pattern** of ice emergence. We are not looking for a perfect ice emergence velocity, that is not possible, but we give the best, defensible estimate of emergence velocity. That pattern is consistent with the rest of the analysis.

L129. I do not think these two cases of bed were described earlier in the text. Why the need for the Flat bed?

We will clarify this. We include a flat bad to show the effect of changing the bed map. And show that the ice emergence rate pattern we highlight is not necessarily dependent on the assumptions of the bed maps from Farinotti.

L155. The fact that debris thicken downglacier is probably repeated close to 10 times in the three papers (and also plot many times). This is irritating. It illustrates why the artificial separation in three papers does not work.

Thank you for pointing out this repetition we are happy to state ' thickening debris downglacier' it less often. But we are not clear how this is the result of artificial separation of the papers. The reviewer could be more clear. We feel that this is just the result of us needing to smooth the writing for this manuscript.

Whole Section 4.1.2. I am not sure I get the point here and I do not really understand what is the actual finding: thick debris are found on almost all stagnating glacier tongues where melt rates are low, emergency velocity and dh/dt also. There is nothing really new here.

Thank you for pointing this out, we are trying to show how the very pattern you describe is then reflected in many other properties and how that is related to Ostrem's curve. We are just laying the foundation for further analysis lower down in the manuscript.

Also I do not understand why the authors consider a steady state to interpret the evolution a glacier that is actually far from equilibrium. How debris are distributed nowaday is probably inherited from decades of imbalance.

We are just saying that a glacier is always trying to balance the effect of low melt (under thick debris) with reduced ice flow even if the glacier is not in steady state. This is a very difficult concept to communicate but it is important for laying the foundation for what is below.

L202-210. I find this part of the text poorly connected to the data/results obtained. Such a discussion would be relevant for a study examining time series of images and able to observe those debris mass wasting events related to the heterogeneity of the melt rate. Right now, no data in the study allow elaborating or confirming such a theory as a one-shot debris thickness and cliff distribution map was produced.

We are outlining a holistic perspective. These are arguments that are outlining the range of controls on ice cliff distribution. We are laying out the process basis for further analyses to follow later. We are surprised at the resistance to us presenting ideas that are based on physical arguments. Not all contributions need to be quantitative. We are crossing over into areas that are closer to hypotheses and we aren't sure how that is not a valid contribution.

Reviewer 4 has a good perspective here:

"I don't think it [this manuscript] provides definitive answers to the problem of quantifying the feedbacks in these complex systems, but it does point to a way forwards." As such we think this is manuscript an important contribution to the rock glacier literature.

L310. Glacier

L311. "an ice cliff-glacier thinning feedback is evident on Kennicott Glacier". This was not evident at all for me, I do not think it was demonstrated or I did not get it.

We are proposing that

Figures 1, 2 are good examples of redundant figures, already shown almost identically in part A and B.

As the writers of these manuscripts we had to make decisions that from our perspective can be viewed as arbitrary from the reader (like where to add streams or which figures to repeat). We suspect that if we didn't include these figures then the reviewer would take issue with having to look back at the other parts to see these keep results. From our view this is a minor amount of repeat. If we published these three manuscripts independently there would be no complaints about repeating a context figure.

Figure 3. Do the authors have evidences of reduced ice fluxes with time? This is probably an important part of the story, it is indicated on this figure but not really in the paper. Are the surface velocities changing with time? Or only the reduction in ice fluxes is due to surface lowering? These changes ice fluxes are probably key to understand the present-day distribution of dh/dt and debris on the tongue.

Thank you for highlighting this. We have extracted additional annual surface velocity patterns from and earlier time period and we will include them with flux estimates from earlier in the next iteration of the manuscript. As we should all expect though (we know ice has thinned, velocity has lowered) but we can show quantitatively that ice flux is reduced.

Figure 5d. The difference between "Flat and Variable" bed needs to be discussed more. It is worrisome that the "Flat Bed" curve show nearly 0 emergence velocity in

region of high melt, in the active ice zone.

The reviewer has misread the plot here, the variable bed plot shows nearly 0 emergence velocity. But the surface velocity pattern in this region is nearly uniform in this region. If the ice thickness is also nearly uniform in this region then ice emergence velocity can also be 0. This is totally physically viable.

**Part C: proposed changes**

We want to emphasize here that we do outline new feedbacks in this paper.

From Reviewer 3 from Part C:

"P 2 line 62-63: importantly in part C you not just present data on ice dynamics and supraglacial streams but crucially in part C these data and all components of the mass conservation equation (thinning, flux divergence. . .) are analysed for relation and feedbacks between them. Also say this here, as it is the backbone and most exiting part of this part C."

On Kennicott Glacier there is a strong correspondence between ice cliffs and active ice flow. While weak relationships have been suggested here on Kennicott the correlation is more clear than anywhere else.

The highest concentration of ice cliffs occurs at the upper end of the zone of maximum thinning. The high concentration of ice cliffs also corresponds to where we expect ice emergence rates to be high. These ice emergence rates uplift the glacier surface, working to counter glacier thinning. But ice dynamics, which produce this surface uplift also seems to produce more ice cliffs (see the physical descriptions within the main article). These ice cliffs counter the effect of surface uplift, they are essentially a negative feedback on the effect of ice dynamics.

In addition to this new feedback we also present a number of new hypotheses for the interaction of surface processes with melt and ice dynamics with a new, holistic perspective.

We feel that there is more than enough new material here for a stand alone paper, but in order to improve the manuscript we propose that we add these additional datasets/ideas to Part C:

- New annual surface velocities from 2000-2010
  - These velocities allow us to calculate changes in ice emergence rate and ice flux over the in situ measurement period
  - More detailed discussion of the reduction of ice emergence rate through time.

- Delineation of drainage basins on the glacier surface (new figure) to support the stream story already within the manuscript.

- Tie in a discussion about glacier surface topography. Ice cliff maximum heights (from in situ measurements), the number of individual ice cliffs with elevation band, and calculated glacier surface relief down glacier.

- New processes drawings to show the important new observations that we are highlighting in this paper. This will greatly improve the reader's ability to see the new process links we are describing.

- Additional photo evidence from the field outlining these new processes links. Many will go into the supplemental but they will support and clarify the process links we are highlighting.

- Description of a new ice cliff burial mechanism. Timelapse movies from the Kennicott and Ngozumpa glaciers (in the supplemental) showing a new mechanism for the burial of ice cliffs. The actual process is not yet described in detail in the text.

- A paragraph that is the same for each of the 3 parts that outlines how they build off of one another.

---

## Author Comment (AC3) · 15 Feb 2020

Thank you kindly for taking the time to review this manuscript!

This manuscript investigates the relation between the pattern of long-term thing of a
debris covered glacier tongue in Alaska and the debris cover, surface features (cliffs,
channels, ponds), flow dynamics. From this, it convincingly identifies and discusses
in detail the emerging feedback between the related processes and quantities and
thereby contributes to the very relevant and important discussion of what the role of
glacier dynamics and surface features for the thinning (surprisingly high) of debris cov-
ered glaciers are. Some new results are also presented in this paper (channel map-
ping, sinuosity. . .) but the strength of this paper is the very systematic analysis and
discussion of the different terms of the continuity equation that determines the thin-
ning of a glacier (see fig 5). The undertaken bulking of the different quantities into few zones
(upper/lower limb ZWT,. . .) in the discussion of the results helps thereby to get a
clearer picture and to identify the most dominant quantities and feedbacks. There are
a few earlier papers available that tried to address the issue of anomalous thinning of
debris covered glaciers but with a different approach (maybe that should be referenced
better) and i think the identified importance of the reduced ice-emergence (reduced dy-
namic replacement) is very well supported by the presented data and analysis. Thus,
overall this study presents a very interesting and important advance in understanding
the dynamics of debris covered glaciers, a topic of high relevance in times of global
warming, and hence this manuscript a very valuable contribution at TC. There are a
few comments or and issues I have with this paper, but they are mostly rather minor
(see list below) but would hopefully further improve an already very interesting, good
quality and exiting paper. The figure and visualization are in general effective and the
paper is well written.

More general comments 1. This paper is the last (part c) of a series of 3 papers, and
one could always ask if not all parts should have been integrated into one paper. I
admit that some repetition (in the description of some the data sets for example) is
unavoidable, but for me part C works very well as a stand-alone paper and has a very
clear own focus on the dynamic feedbacks and interactions and more than enough
conclusive results for a stand-alone paper. I have to note that I only briefly looked into
the other two papers (part A and B) but it was clear that there the main aim and focus
of the otehr parts (A and B) were substantially different and in my view justified as
separate papers. Moreover I believe that the main messages and findings of the three
papers come in separate papers probably better across than in one huge one.

Thank you kindly. If we do not reply to the comment below we will enact the change in Part C.

2. Abstract focus: somewhat related to the 3-part paper thing, when reading the ab-
stract I got the impression that the main focus of the paper on the feedbacks and inter-
relations between processes to explain the thinning pattern is rather thinly represented

within the abstract (last 3 lines, and little on feedbacks but rather on correlations) and the results used in this papers but from part A and B get too much space in the abstract. A better balance and more focus on the feedbacks and findings of THIS part C would be useful.

We will follow this advice.

3. Difference in time periods of datasets: One potential criticism of the analysis and conclusions one could have is that the thinning-data represents an average over several decades whereas the velocities and surface features, debris extent etc are the 'now' situation. I myself do not really think this is really a big issue but some more explanation and justification for this maybe useful.

We have additional dh/dt data that covers the in situ measurement time period.

4. Literature: With regard to influence/link of ice dynamics to thinning, debris cover and ice cliffs (e.g. explaining anomalous thinning) the paper by Banjeree (2017, TC), Rounce et al (2017), Ragettli (2016, TC) and potentially Moelg et al. (2019, TC) maybe useful to be considered.

Banerjee A. (2017): Thinning of debris-covered and debris-free glaciers in a warming climate. Brief communication. The Cryosphere, 11, 133-138, 2017 www.the-cryosphere.net/11/133/2017/ doi:10.5194/tc-11-133-2017

Rounce, D. R., King, O., McCarthy, M., Shean, D. E., & Salerno, F. (2018). Quantifying debris thickness of debris-covered glaciers in the Everest region of Nepal through inversion of a sub-debris melt model. Journal of Geophysical Research: Earth Surface,123, 1094-1115. https://doi.org/10.1029/2017JF004395

Ragettli, S., Bolch, T., & Pellicciotti, F. (2016). Heterogeneous glacier thinning patterns over the last 40 years in Langtang Himal, Nepal. The Cryosphere, 10(5), 2075-2097. https://doi.org/10.5194/tc-10-2075-2016

Moelg N., T. Bolch, A. Walter and A. Vieli (2019) Unravelling the evolution of Zmuttgletscher and its debris cover since the end of the Little Ice Age. The Cryosphere, 13, 1889-1909, https://doi.org/10.5194/tc-13-1889-2019

Thanks we will include these citations.

Minor/specific comments

p. 1 Line 15: the term 'melt hotspots' is here not really clear maybe specify a bit more what it is ('melt hotspots such as ice cliffs or channels')

p 1 lines 23-27: maybe make clearer what of these results from this part C paper and what is from earlier (or really focus on part C part).

P 1 line 24-25: high melt and HIGH melt gradients? here and also on next line it is not so clear to me what you mean by 'melt gradients' here, 'spatial gradients in melt' along flow, gradients in melt with regard to changing debris thickness. . .. be clearer.

P1 line 31: a brief explanation why ice cliffs are most abundant at the upglacier end

maybe useful here (I think you have some idea about this or am I wrong?).

P 2 line 50: I think 'surface' uplift is here not quite correct, ice emergence is the relative movement to the surface or particle uplift against the surface, so maybe 'ice' uplift is more appropriate.

Thanks for clarifying this for us.

P 2 line 56-57: '. . .will facilitate the INTERPRETATION AND prediction of. . .'

P 2 line 62-63: importantly in part C you not just present data on ice dynamics and supraglacial streams but crucially in part C these data and all components of the mass conservation equation (thinning, flluy divergence. . .) are analysed for relation and feedbacks between them. Also say this here, as it is the backbone and most exiting part of this part C.

Thanks for this.

p. 3 lines 73-79: is this paragraph on the water pressure variations and sliding really needed? Maybe just summarize it in one sentence in the section 2.1 or 3.1 on the velocity data.

We will clarify this and make the connections more clear.

p 4 line 116: what is grid size chosen?

We will mention the grid size here.

p. 5 lines 143-153: maybe this paragraph (together with next paragraph) can be shortened a little bit as already presented in part A and B.

p. 5 line 165: be clearer here on: 'THE ALONG FLOW/PROFILE PATTERN OF annual surface velocities. . ..

p. 6 line 171: this link between debris thickness and flow dynamics is a consequence of the continuity equation, so maybe be more explicit on this. '. . .controls the melt rate and which a consequence of mass continuity is linked to ice dynamics.

p. 6 line 174-176: this complementation of debris thickness melt rates and surface velocity probably is meant in a steady state sense, otherwise the dhdebris/dt should also be mentioned (maybe clarify). Further to this sentence, with patterns I assume SPATIAL (along flow) PATTERNS are meant?

We will clarify.

p. 6 line 183: again, '. . .SPATIAL /ALONG FLOW gradients in melt are low. . .'

p. 6 lines 186-191: good point!

p. 6 line 197-198: maybe explain why streams disappear in lower limb, is it because the drain through moulins in transition, but why are moulins there, connection to strain rates (longitudinal stretching? Not so clear in Fig. 5. )

We will explain this.

p. 7 line 234: should it not be 'The lowest 4 km of the glacier ARE. . ..' (it is 4). And again why is this disconnection there, because water is drained though moulins to bed. . ..

p. 8 line 255. Maybe refer to Fig. 5b+f after (ZMT). Further with 'changes in ice flow' you probably mean along flow changes in ice flow, or more specifically the flux divergence or emergence rate.

p. 8 line 251: 'Process links . . .'

p. 9 line 273: not sure why you are so vague in your statement herewith 'may' be related. Why not be a bit more direct and say 'seems related to. . .'. Further do you mean to the 'SPATIAL/ALONG FLOW pattern'?

p. 9 line 290: 'increased ice strain', I struggle to see longitudinal extension (strain)here at the transition from the upper to the lower limb, the velocities clearly decrease down glacier there, so it would rather mean 'compression' or do I get something wrong here?

Fig. 1: the label dH/dt in the figure is rather confusing to read, make sure all is on one line e.g. dH*dt^(-1). Caption line 6: do you mean opposite 'sign' rather than 'sight'?

Caption line 7: 'The black line shows the profile used . . ..'

Fig. 2: I know that the exact threshold between upper and lower limb is not crucial but in the text a rough transition between 10cm and 20cm is given why not indicate this in the figure maybe as a grey shaded bar in the background.

We will.

Fig. 3: caption line one I would add at end of first sentence '. . . along the profile indicated in fig. 1c.'

Fig. 4: a detail but the map could do with a scale. More importantly, where in the figure/map is the box indicating the extent of Figure 11? I can simply not find it.

Fig. 5: in sub-fig (f) and in caption line 8, strictly speaking the label should be 'elevation change rate' as the sign is already negative and a lowering rate that is negative would then mean thickening again. Caption line 1: again make clear that the show data are '. . . for the swath along the profile indicated in fig. 1c.'

We will change this.

Fig. 7: here an elevation threshold/bands are used to summarize/group the sinuosity data, but am I right that these are both above the ZMT and in the upper limb of the oestroem curve. This maybe useful to be explained in the caption.

We will change this.

Fig. 10. I found this figure rather difficult to read, there is a lot of information and detail and I initially expected from this schematic to better get the big picture. Maybe I just

expected the wrong thing and the colors (blue or red) were not so clear to me and I wondered if it really helped me a lot. If I see it as complete documentation of all different relations and feed backs it is maybe fine, but then maybe it should be phrased as such. More importantly, in the caption the colors red and blue refer to positive effects or negative effects but it is not so obvious to me what you mean by positive and negative. Does this refer to positive and negative FEEDBACKS (self enhancing/reducing) or positive/negative from a glacier health (negative mass loss, reduced speed,. . ..). should be clarified.

We will clarify this figure or remove it.

**Part C: proposed changes**

We want to emphasize here that we do outline new feedbacks in this paper.

From Reviewer 3 from Part C:

"P 2 line 62-63: importantly in part C you not just present data on ice dynamics and supraglacial streams but crucially in part C these data and all components of the mass conservation equation (thinning, flux divergence. . .) are analysed for relation and feedbacks between them. Also say this here, as it is the backbone and most exiting part of this part C."

On Kennicott Glacier there is a strong correspondence between ice cliffs and active ice flow. While weak relationships have been suggested here on Kennicott the correlation is more clear than anywhere else.

The highest concentration of ice cliffs occurs at the upper end of the zone of maximum thinning. The high concentration of ice cliffs also corresponds to where we expect ice emergence rates to be high. These ice emergence rates uplift the glacier surface, working to counter glacier thinning. But ice dynamics, which produce this surface uplift also seems to produce more ice cliffs (see the physical descriptions within the main article). These ice cliffs counter the effect of surface uplift, they are essentially a negative feedback on the effect of ice dynamics.

In addition to this new feedback we also present a number of new hypotheses for the interaction of surface processes with melt and ice dynamics with a new, holistic perspective.

We feel that there is more than enough new material here for a stand alone paper, but in order to improve the manuscript we propose that we add these additional datasets/ideas to Part C:

- New annual surface velocities from 2000-2010
  - These velocities allow us to calculate changes in ice emergence rate and ice flux over the in situ measurement period
  - More detailed discussion of the reduction of ice emergence rate through time.

- Delineation of drainage basins on the glacier surface (new figure) to support the stream story already within the manuscript.

- Tie in a discussion about glacier surface topography. Ice cliff maximum heights (from in situ measurements), the number of individual ice cliffs with elevation band, and calculated glacier surface relief down glacier.

- New processes drawings to show the important new observations that we are highlighting in this paper. This will greatly improve the reader's ability to see the new process links we are describing.

- Additional photo evidence from the field outlining these new processes links. Many will go into the supplemental but they will support and clarify the process links we are highlighting.

- Description of a new ice cliff burial mechanism. Timelapse movies from the Kennicott and Ngozumpa glaciers (in the supplemental) showing a new mechanism for the burial of ice cliffs. The actual process is not yet described in detail in the text.

- A paragraph that is the same for each of the 3 parts that outlines how they build off of one another.

---

## Author Comment (AC4) · 15 Feb 2020

Reply to Reviewer 4 Part C

Martin Kirkbride (Referee)
m.p.kirkbride@dundee.ac.uk

Thank you kindly for taking the time to review this manuscript! We appreciate it.

This interesting and topical paper synthesizes a range of glaciological data to improve understanding of the process feedbacks between glacier flow, melt distribution under debris cover, and thinning, at a large compound Alaskan glacier. The ambition of the paper is welcome: there is an increasing output of papers dealing with one or two aspects of debris-covered glacier (DCG) monitoring and evolution, many based on state-of-the-art data gathering, but few attempts have hitherto been made to understand interactions at appropriate timescales, and to come up with integrative explanatory models. The paper bases its approach on mass continuity and the debris-thickness/melt relationship (the Ostrem Curve).

Thank you kindly for taking the time to review the manuscript!

My general comment is that this is a rigorous and well-argued study which shows some interesting results, different from other papers I am familiar with. The core finding is that the interaction of ice flow, debris emergence, melt and thinning have produced a subtle "bulge" several kilometres above the terminus, marking the transition from active ice flow and debris emergence upstream to relatively stagnant, heavily debris-covered ice downstream. My surprise is that the active/stagnant transition is manifest as a convexity in the long profile, rather than a concavity as described in DCGs elsewhere. While Figs 2 and 5 are vertically exaggerated to show this subtle topographic evolution (as they must be), it is convincingly demonstrated. The transition corresponds to the kink in the downward limb of the Ostrem Curve at which the rate of sub-debris melting becomes less sensitive to debris thickness.

The paper raises some interesting questions, but also contains some inferences of cause-effect which are less well substantiated than others. There is perhaps a tendency in places to make easy inferences of causation based on only the available data, when other variables have not been considered. (This is not to denigrate the high-quality datasets presented). As such, I don't think it provides definitive answers to the problem of quantifying the feedbacks in these complex systems, but it does point to a way forwards.

We appreciate you bringing this issue to light. We agree that some of the inferences are just hypotheses that need to be tested and are better supported than others. We will make sure we indicate more clearly where these hypotheses are less well founded.

Another issue (also in no way a criticism here) is that the literature presents the "debris-covered glacier" as if it is a single class of glacier: this is not the case. DCGs take many forms and origins, and are unlikely to have a single unifying model of behaviour and evolution. This study of Kennicott Glacier is of a very large compound valley glacier terminating in a proglacial lake, whose debris cover is fed by coalescing medial moraines. We might not expect models from this glacier to apply easily to (for example) smaller moraine-dammed DCGs whose flow is obstructed towards the terminus, or single-basin glaciers with transverse foliation. Perhaps some acknowledgement of this diversity would be appropriate.

Thank you for raising this point. We agree that debris-covered glaciers take very diverse forms. We do feel though that it is something of an open question what about differences between these glaciers really matters. We will address this issue in the manuscript.

It isn't clear from this paper (Part C of three) what the ice thickness distribution is, but this information would be useful.

We will be sure to include the ice thickness estimates in the supplemental of Part C.

This is because, while velocity evolution is a key variable, the causes of velocity change and its distribution on the long profile are not covered, yet this information is essential for understanding the dynamic evolution of the glacier. I would like to see some consideration of the effects of both thinning rates and surface gradient changes on the driving stresses, to explore why the observed pattern of stagnation has developed: it implies a collapse in the driving stress from the terminus upstream, which in turn must be some combination of reduced ice thickness and slope. It is noteworthy (though largely unrecognised generally) that very thick, very gentle glaciers such as DCG tongues are sensitive to small changes in slope, at least as much as in thickness. So there is scope for a fuller explanation than is given in the manuscript.

Thank you kindly for this discussion. We agree and we have now extracted earlier annual surface velocities so we also add those to this manuscript. We will further address the topics discussed in this paragraph in the revised manuscript.

I have some minor line-by-line comments to improve the presentation, and to correct minor editorial mistakes (attached). (If we do not respond to the minor comment we will make the necessary change.)

Line 15 How can mass balance be "enhanced": rephrase.

Need to define upper limb and lower limb of Østrem's curve, because what is referred to here are really segments of the same limb (debris thicker than effective thickness). Don't hyphenate "upper limb" or "lower limb".

We will clarify this.

Suggest "in spite of" instead of "as well as"?

"may...control": it clearly does!

36-7 Why is the term "melt hotspots" in italics? Unnecessary.

Although the term "debris-cover anomaly" has gained currency since 2015, there is often a careless use of terminology in this context, where glacier thinning and melting are used synonymously. The "anomaly" (if one exists" is in the thinning rates, not the sub-debris melt rates. Make this clear.

We agree that the 'DC anomaly' refers to thinning rate not melt rates. This distinction is key for appreciating the three papers we lay out here.

"causes", not "cause" (process is singular).

62-3 One cannot estimate a supraglacial stream. Rephrase.

"south-facing"

Suggest "more cliffs per unit area".

Add comma after "significantly".

Add hyphen after "column".

Remove hyphen after "corrected".

I take issue with the use of "bi-modal" here, because a bimodal distribution has two modes (peaks). Here, the term is used to indicate an absence of streams on thickly debris-covered ice: this isn't "bimodal", rather it's a threshold control.

We will rephrase this.

147-152 Remove hyphens in "upper limb" and "lower limb". See comment re. line 24 about clarity of what these terms mean.

Remove italics: unnecessary.

See l. 147

Why use the term "attractor state" here? You imply the glacier is attracted to an equilibrium state of mass balance, but there's no reason for this to be more likely than any other mass balance state because mass balance is not controlled by internal system dynamics.

We need to clarify this. What we mean is that changes in surface mass balance translate into changes in ice dynamics they feedback into one another. We will work to make this more clear for the reader.

Commas after "are high" and "are low".

Re. chicken-egg quandary: this disappears if a longer-term view is taken, in which velocity is the ultimate control, because the glacier must slow down to allow debris cover to accumulate ("ablation-dominant" conditions of Kirkbride (2000 IAHS))". So the question becomes what causes change to the longitudinal velocity profile of the glacier over time, where does velocity reduce earliest on this profile, and why? (See my general comments).

Great topic, worth further exploration outside of this manuscript though.

See l. 135

See l. 135

et seq. It's really no surprise that streams are more abundant on steeper gradients, and lakes on gentle gradients, since water flows downhill. What point is being made here?

We are just setting up the observations of surface features related to Ostrem's curve so we can discuss them below. We will ad text so the reader feels more grounded in revisions.

I'm perplexed by the conclusion that ice cliff abundance is related to basal sliding rate. I simply don't see a direct connection here, and wonder whether you are taking spatial associations too far down the line of causal relationships. If the connection is indirect, it needs to explained clearly and in full.

Here is a place where we need to be more clear about the potential connection. In the summer basal sliding produces high gradients in glacier surface velocity. It may be that high gradients of surface velocity disturb debris-covered slopes leading of the failure of debris and the exposure of ice which can then become ice cliffs.

I don't understand how stream undercutting od ice walls increases debris thickness at the base of the ice slope. This implies that the ice slopes must decline in angle, for which no evidence is given: parallel retreat will give the same thickness at the base as at the top. (More likely, fluvial removal ofdebris occurs, so an apparent thickening as seen in Fig 9 may be debris brought to the site from upstream). Suggest omitting these two sentences.

Again we need to be more clear. Often times, and we will highlight this with photo evidence, sinuous streams persist at the base of ice cliffs. These sinuous streams tend to produce ice cliffs that have an arc shape. The arcuate planform then tends to focus debris at its base. Furthermore, sometimes ice cliffs have drainage basins on their surface which we observed to also funnel debris towards their base.

I disagree that the lower glacier is "hydrologically disconnected". Supraglacial drainage becomes englacial (and subglacial?) which isn't the same as being disconnected (see Fyffe et al 2019 J Hydrol 570, 584-597).

We can rephrase this such that the 'surface hydrology is disconnected.' We observed the termination of surface streams in a chain at this transition. The digitization of the streams highlights this.

"Ice cliffs are ... more likely to be buried". Buried how? This assumes a process of disappearance which isn't explained. I agree with the general point about their removal, but the process needs careful explanation.

Yes we will explain it in more detail as per many of the other processes.

Debris cover and surface drainage basin relationships are shown nicely in Catriona Fyffe's recent paper (see l. 234 comment).

We will be sure to cite this paper. We may also have time to produce a similar surface drainage map as well for Kennicott Glacier.

The effect of this slope reduction is probably a key observation, because on thick, gentle glaciers the driving stress can be at least as sensitive to small changes in slope as to ice thinning. It would be interesting to see how this slope reduction plays out with changes to the basal stress profile over time, which may show something useful re. Velocity.

Great idea we will look into this.

See l. 242: on steep, thin glaciers, thickness change is the main control: on gentle, thick glaciers, slope is more important. Perhaps refine this sentence in the context that DCGs are characteristically thick and gentle.

Yes we will do this.

Replace "pattern of debris" with "distribution of debris thickness": be specific.

"... this pattern over time"

Desperately needs a comma after " thinned" , otherwise the sentence makes no sense.

"have", not "has". The stated change in the surface flow field is not supported by any evidence. Either omit this point, or provide evidence for it. If true (which I'm sure it is), clean ice would be redistributed as well as debris-covered ice, so is it an explanation at all?

This is an interesting point. We will think more about how to explain this. We are convinced based on the mass conservation equation for debris that surface velocity gradients are key for velocity change. We will highlight this effect based on this equation.

See l.147

Spelling "Glaicer"

"DS" is acknowleged here, but isn't a named author of the paper.

Captions

Fig 1 Panel (a) doesn't show the location of Panel (b).

Fig 2. I would go further in saying the elbow of the curve lies between 12 and 14 cm. Could you fit best-fit lines to each segment iteratively to find the location of the angle? Also, highlight the bare ice point more clearly. Which altitude does this point originate from? (it can't be a unique point).

We will better clarify the bend we refer to.

Fig 5. The key figure in the paper, and really interesting to absorb. One query is why in (e) the elevation difference decreases below c. 3km above the terminus, but in (f) the surface lowering rate increases over the same distance? This seems inconsistent.

Hmmm. Just how the data work out. We will confirm that the origin of each data profile is the same, but it is likely just the result of the data.

Fig 7. While interesting in its own right, I'm sure what data on stream sinuousity contributes to the overall interpretations and conclusions. This figure and the accompanying text could be omitted, unless a stronger case is made for its inclusion.

We will better explain this effect. The more sinuous a stream the more length of ice cliff the stream can undercut. The more sinuous the stream the more ice cliffs will be arc shaped and the more likely these ice cliffs are to focus debris at their base.

**Part C: proposed changes**

We want to emphasize here that we do outline new feedbacks in this paper.

From Reviewer 3 from Part C:

"P 2 line 62-63: importantly in part C you not just present data on ice dynamics and supraglacial streams but crucially in part C these data and all components of the mass conservation equation (thinning, flux divergence. . .) are analysed for relation and feed-backs between them. Also say this here, as it is the backbone and most exiting part of this part C."

On Kennicott Glacier there is a strong correspondence between ice cliffs and active ice flow. While weak relationships have been suggested here on Kennicott the correlation is more clear than anywhere else.

The highest concentration of ice cliffs occurs at the upper end of the zone of maximum thinning. The high concentration of ice cliffs also corresponds to where we expect ice emergence rates to be high. These ice emergence rates uplift the glacier surface, working to counter glacier thinning. But ice dynamics, which produce this surface uplift also seems to produce more ice cliffs (see the physical descriptions within the main article). These ice cliffs counter the effect of surface uplift, they are essentially a negative feedback on the effect of ice dynamics.

In addition to this new feedback we also present a number of new hypotheses for the interaction of surface processes with melt and ice dynamics with a new, holistic perspective.

We feel that there is more than enough new material here for a stand alone paper, but in order to improve the manuscript we propose that we add these additional datasets/ideas to Part C:

- New annual surface velocities from 2000-2010
  - These velocities allow us to calculate changes in ice emergence rate and ice flux over the in situ measurement period
  - More detailed discussion of the reduction of ice emergence rate through time.

- Delineation of drainage basins on the glacier surface (new figure) to support the stream story already within the manuscript.

- Tie in a discussion about glacier surface topography. Ice cliff maximum heights (from in situ measurements), the number of individual ice cliffs with elevation band, and calculated glacier surface relief down glacier.

- New processes drawings to show the important new observations that we are highlighting in this paper. This will greatly improve the reader's ability to see the new process links we are describing.

- Additional photo evidence from the field outlining these new processes links. Many will go into the supplemental but they will support and clarify the process links we are highlighting.

- Description of a new ice cliff burial mechanism. Timelapse movies from the Kennicott and Ngozumpa glaciers (in the supplemental) showing a new mechanism for the burial of ice cliffs. The actual process is not yet described in detail in the text.

- A paragraph that is the same for each of the 3 parts that outlines how they build off of one another.